# SWITCH 1/DYAD is a WINGS APART-LIKE antagonist that maintains sister chromatid cohesion in meiosis

Chao Yang [1], Yuki Hamamura[1], Kostika Sofroni [1], Franziska Böwer[1], Sara Christina Stolze [2], Hirofumi Nakagami [2] & Arp Schnittger [1]

Mitosis and meiosis both rely on cohesin, which embraces the sister chromatids and plays a crucial role for the faithful distribution of chromosomes to daughter cells. Prior to the cleavage by Separase at anaphase onset, cohesin is largely removed from chromosomes by the non-proteolytic action of WINGS APART-LIKE (WAPL), a mechanism referred to as the prophase pathway. To prevent the premature loss of sister chromatid cohesion, WAPL is inhibited in early mitosis by Sororin. However, Sororin homologs have only been found to function as WAPL inhibitors during mitosis in vertebrates and *Drosophila*. Here we show that SWITCH 1/DYAD defines a WAPL antagonist that acts in meiosis of *Arabidopsis*. Crucially, SWI1 becomes dispensable for sister chromatid cohesion in the absence of *WAPL*. Despite the lack of any sequence similarities, we found that SWI1 is regulated and functions in a similar manner as Sororin hence likely representing a case of convergent molecular evolution across the eukaryotic kingdom.

[1] Department of Developmental Biology, University of Hamburg, Hamburg 22609, Germany. [2] Max-Planck-Institute for Plant Breeding Research, Cologne 50829, Germany. Correspondence and requests for materials should be addressed to A.S. (email: arp.schnittger@uni-hamburg.de)

The tight regulation of sister chromatid cohesion is essential for accurate chromosome segregation during mitosis and meiosis. During S-phase, the genomic DNA is duplicated resulting in the formation of two sister chromatids per chromosomes. The newly formed sister chromatids are held together by the cohesin complex, which builds a ring-like structure embracing the chromatids. Besides sister chromatid cohesion, the cohesin complex is crucial for genome stability, DNA repair, chromatin structure organization, and gene expression[1–4].

The cohesin complex is highly conserved in the eukaryotic kingdom with homologs present from animals to plants comprising four core subunits: SMC1 and SMC3, two ATPases that belong to the family of structural maintenance of chromosomes (SMC) proteins, the heat-repeat domain protein SCC3/SA and one α-kleisin component RAD21/SCC1, which is replaced in meiosis by REC8/SYN1.

The presence of cohesin on chromosomes is very dynamic. Cohesin is already loaded onto chromosomes by the SCC2-SCC4 loader complex during the G1 phase of the cell cycle. Sister chromatid cohesion is established in the subsequent S-phase and regulated by several cohesin accessory proteins, including the PRECOCIOUS DISSOCIATION OF SISTER 5 (PDS5) and WINGS APART-LIKE (WAPL)[5–7]. PDS5 assists the acetylation of the SMC3 subunit by Establishment of cohesion 1 (Eco1)/Chromosome Transmission Fidelity 7 (CTF7), needed to close the cohesin ring[8–10]. Cohesin is then maintained on chromosomes until late G2 in the mitotic cell cycle and early prophase I in meiosis, respectively. As cell division is approaching metaphase, cohesin, especially on chromosome arms, undergoes tremendous removal mediated by the cohesin dissociation factor WAPL, a process known as prophase pathway of cohesin removal[11–14]. At the centromeric regions, cohesin is largely protected by the Shugoshin-PP2A complex[15,16]. This centromeric cohesin is released by a Separase-dependent proteolytic cleavage of the kleisin subunit RAD21/REC8, thereby allowing the separation of sister chromatids at anaphase onset (anaphase II in meiosis).

To prevent a premature release of sister chromatid cohesion in mitosis, especially on chromosome arms, Sororin counteracts the releasing force of WAPL by binding to PDS5 and displacing WAPL from PDS5[11,17–19]. However, Sororin has so far only been identified in vertebrates. More recently, an ortholog of Sororin, named Dalmatian, was found in Drosophila, which exert both Sororin's cohesin stabilizing and Shugoshin's cohesin protecting functions in mitosis[20].

In late prophase, Sororin is recognized by the APC/C^Cdh1 (Anaphase-promoting complex/cyclosome) and degraded by the ubiquitin–proteasome pathway, thereby releasing its repression of WAPL and activating the prophase removal of cohesin[18,21]. Phosphorylation through Cdk1 (cyclin-dependent kinase 1) and Aurora B kinase serves thereby as a signal for the degradation of Sororin[22,23].

In contrast to mitosis, it is not clear how sister chromatid cohesion is protected during early meiotic prophase I. Notably, Sororin does not seem to play a role for the regulation of meiotic cohesion. Although Sororin is present in male meiosis in mouse, it is exclusively localized on the central regions of the synaptonemal complex (SC) and not on the axial/lateral elements of SC where the cohesin complex is found[24]. This localization pattern makes Sororin unlikely, at least in mouse, be the protector of cohesin. This conclusion is substantiated by the finding that the localization of Sororin in the central region of the SC is not dependent on the meiosis-specific subunits REC8 and SMC1β[24].

In contrast, WAPL has been found to remove meiotic cohesin at late prophase in most if not all organisms studied including Arabidopsis and other plants[11,14,25–27]. Thus, it remains a puzzle how the activity of WAPL is inhibited in early meiotic prophase I especially since no obvious sequence homolog of Sororin or

Dalmatian has been identified in the plant lineage and other major branches of the eukaryotic kingdom[28].

Here, we report that the previously identified SWI1 gene in Arabidopsis encodes a WAPL inhibitor. Despite any sequence similarities between SWI1 and Sororin, we further reveal that SWI1 antagonizes WAPL in prophase I of meiosis through a similar strategy as Sororin in mitosis. Moreover, SWI1 turned out to be amazingly similarly regulated in Arabidopsis as Sororin in vertebrates.

## Results

**Meiotic cohesin removal is mediated to large extent by WAPL.** To get an understanding of cohesin dynamics during meiosis, we followed the expression and localization of a previously generated functional REC8-GFP reporter in male meiocytes by live cell imaging[29]. We observed that the majority of cohesin (~90%) in the wildtype, but not in the previously described wapl1 wapl2 double mutant[11], is already largely released from chromatin prior to anaphase I indicating that the impact of the WAPL-dependent prophase pathway on cohesin removal is very strong in male meiosis of Arabidopsis (Fig. 1a–c; Supplementary Movies 1 and 2).

To follow WAPL1, we generated a WAPL1-GFP reporter, which fully complemented the wapl1 wapl2 defects (Supplementary Fig. 1) and accumulated in somatic cells of the anther and in male meiocytes. In meiocytes, the WAPL1-GFP signal showed a homogeneous distribution in the nucleoplasm from pre-meiosis until leptotene, suggesting no or only a very weak interaction of WAPL1 with chromatin (Fig. 1di, ii). Subsequently, foci and/or short stretches of WAPL1-GFP appeared in the nucleus at late leptotene/early zygotene, coinciding with the eviction of cohesin from chromatin (Fig. 1diii). The accumulation of WAPL1-GFP signal on chromatin became more prominent in zygotene and pachytene, which is consistent with the progressive release of cohesin (Fig. 1c, d iv,v). In metaphase I, WAPL1-GFP was found at condensed chromosomes (Fig. 1dvi). While WAPL1-GFP signal is still present in the nucleus after the first meiotic division until tetrad stage, it was not localized to chromatin any longer (Fig. 1dvii, viii). This localization pattern was confirmed by immuno-localization of WAPL1-GFP using an antibody against GFP (Supplementary Fig. 1c).

**SWI1 is expressed in early meiosis.** The observation that WAPL1 is already present in early prophase at a time point when REC8 removal from chromatin has not started, suggested the existence of a WAPL repressor that might prevent WAPL from localizing to chromatin and unloading cohesin prematurely. However, no obvious sequence homolog of Sororin, the only known WAPL repressor in mitosis, exists in Arabidopsis[28]. We reasoned that a potential repressor of WAPL during meiosis should have all or at least some of the following characteristics: first, mutants of this repressor should experience premature loss of sister chromatid cohesion and hence probably have a strong mutant phenotype in meiosis. In turn, this makes it likely that such a mutant has already been identified due to the extensive search for meiotic mutants in Arabidopsis. Second, this repressor would probably be a protein of unknown molecular function. Third, as a regulator of sister chromatid cohesion, this factor should interact with the cohesin complex and hence, its correct localization to chromatin may also depend on a functional cohesin complex.

The gene SWITCH1 (SWI1), also known as DYAD, was previously identified based on its requirement for sister chromatid cohesion in meiosis[30–32]. SWI1 encodes for a protein of unknown biochemical function and its mechanism of action

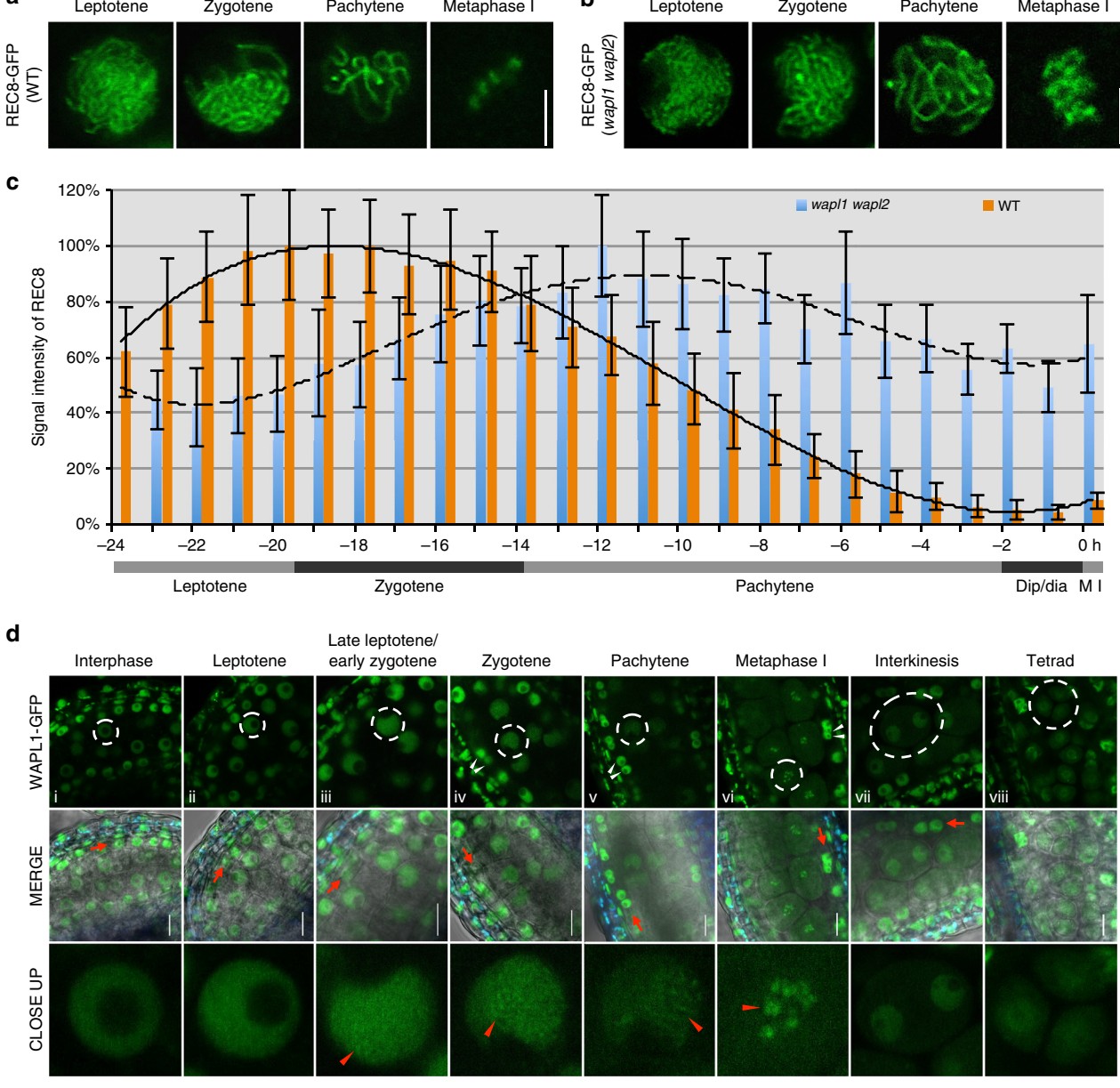

**Fig. 1** Dynamics of REC8 and WAPL in male meiocytes. **a**, **b** Confocal laser scanning micrographs of REC8-GFP localization in male meiocytes in the wildtype (WT) (**a**) and in *wapl1 wapl2* double mutants (**b**). Bar: 5 μm. **c** Quantification of cohesin during meiosis I in male meiocytes of the wildtype (WT) and *wapl1 wapl2* mutants based on a REC8-GFP reporter. The graph represents the relative fluorescence intensity of the REC8-GFP signal; error bar represents standard deviation of at least 10 meiocytes analyzed. Dip/dia diplotene/diakinesis, M I metaphase I. Polynomial trendlines are shown (correlation coefficient $R^2 = 0.997$ and $0.898$ for the wildtype (solid line) and *wapl1 wapl2* (dashed line), respectively. The source data of this graph are provided in the Source Data file. **d** Confocal laser scanning micrographs of WAPL1-GFP in anthers of *wapl1 wpal2* double mutants. Dashed white cycles indicate the meiocytes magnified in the close-up panel in the bottom row. Red arrowheads denote the accumulated WAPL1-GFP signal at chromatin. Red arrows indicate the layer of tapetal cells that are used as one of the criteria for staging. White arrowheads depict bi-nuclear tapetal cells. Bar: 10 μm

has been unresolved up to now. However, SWI1 was previously reported to be exclusively expressed in interphase prior to meiosis and could neither be detected in leptotene nor in any subsequent meiotic stage[30,31]. This expression pattern is difficult to reconcile with the *swi1* mutant phenotype, e.g., a failure to assemble the chromosome axis and to establish sister chromatid cohesion. Therefore, we revisited the expression pattern of *SWI1* in both male and female meiocytes by generating a genomic reporter in which the coding region of *GFP* was inserted directly before the STOP codon of *SWI1*. Expression of this reporter in *swi1* mutants could fully restore a wild-type meiotic program (Supplementary

Fig. 2). To stage the expression of SWI1, we also generated a functional reporter line for the chromosome axis protein ASYNAPTIC 3 (ASY3), where RFP was used as a fluorescent protein (Supplementary Fig. 3).

Consistent with previous reports, SWI1 was first detected as numerous foci/short stretches in interphase nuclei of both male and female meiocytes (Fig. 2a; Supplementary Fig. 4). In addition, the SWI1-GFP signal was present in leptotene and became even stronger as cells progressed through leptotene as staged by the migration of the nucleolus to one side of the nucleus[33–35] and the appearance of an ASY3 signal on condensing chromosomes

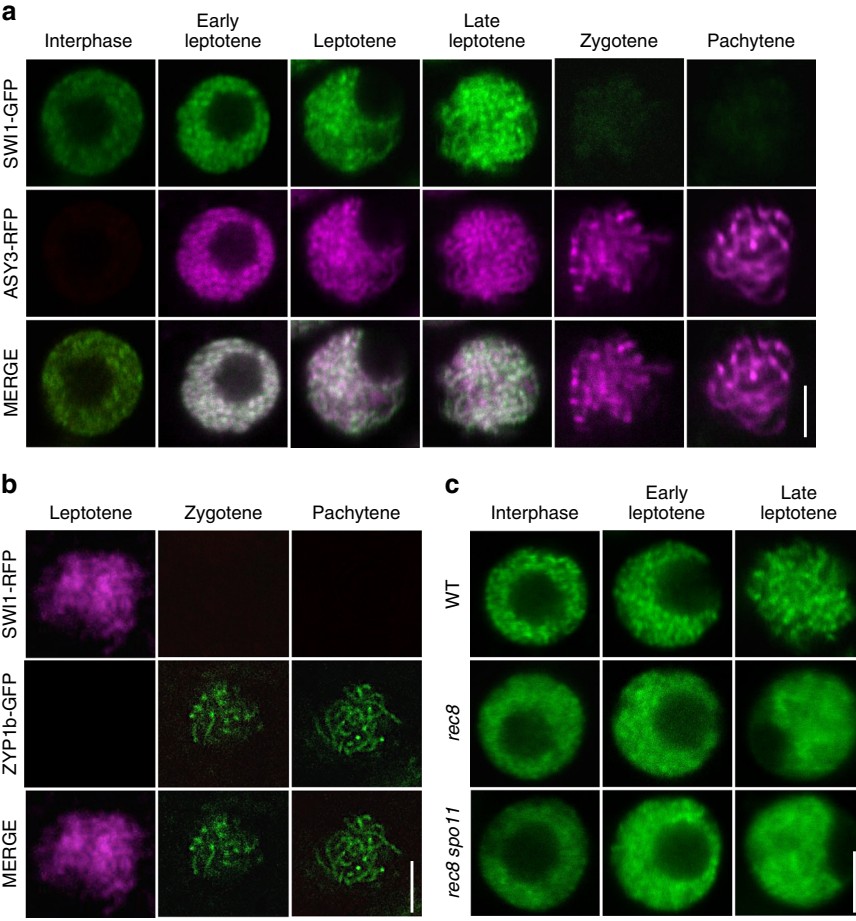

**Fig. 2** Localization pattern of SWI1. Co-localization analysis of SWI1-GFP with ASY3-RFP (**a**) and SWI1-RFP with ZYP1b-GFP (**b**) during interphase and prophase I of wild-type male meiocytes using confocal scanning laser microscopy. **c** SWI1-GFP in the male meiocytes of the wildtype (WT), *rec8* and *rec8 spo11* mutants during interphase and prophase I. Bar: 5 μm

(Fig. 2a; Supplementary Fig. 4). This analysis also showed that SWI1 is chromatin associated. In zygotene, when chromosomes further condensed, highlighted by ASY3-RFP, the SWI1 signal strongly declined until it was not detectable any longer in late pachytene (Fig. 2a; Supplementary Fig. 4).

To confirm that SWI1 reaches its expression peak in late leptotene and decreases by zygotene, we constructed a reporter line for ZYP1b, a component of the central element of the synaptonemal complex. Since a fusion of ZYP1b to RFP resulted in only a very weak fluorescent signal, we generated a ZYP1b-GFP fusion along with a fusion of SWI1 to RFP, which could also restore full fertility and meiotic progression of *swi1* mutants (Supplementary Figs. 5 and 6). In late leptotene, the SWI1-RFP signal is strongly present on chromosomes while no signal for ZYP1b was detected (Fig. 2b). From zygotene onwards, when short stretches of ZYP1b indicate partially synapsed chromosomes, the SWI1 signal was hardly detectable, corroborating that SWI1 is largely absent from chromosomes after zygotene corresponding to the removal of REC8 (Fig. 1a).

**Chromatin association of SWI1 and REC8 is mutually dependent**. Establishment of sister chromatid cohesion has been shown to be compromised during meiosis in *swi1* and cohesin components, e.g., REC8 and SMC3, were found to be not properly bound to chromosomes in this mutant[30]. Using live cell imaging and immuno detection assays, we confirmed these cohesion defects by studying REC8-GFP in three different mutant

alleles, *swi1-2*, *swi1-3*, and *swi1-4*, that showed identical REC-GFP localization defects (Fig. 5a; Supplementary Fig. 7).

To address whether SWI1 localization also depends on cohesin, we introgressed the *SWI1-GFP* reporter into *rec8* mutants. Although no obvious differences were found in interphase in comparison to *swi1* mutants complemented by the expression of *SWI1-GFP*, we found that SWI1 did not properly localize to chromatin in *rec8* mutants in prophase (Fig. 2c). This failure was not due to chromatin fragmentation present in *rec8* since we observed the same pattern when the *SWI1* reporter was introgressed into *rec8 spo11* double mutants in which the endonuclease SPORULATION DEFECTIVE 11 (SPO11) is not functional and hence no double strand breaks are formed.

However, immuno-localization experiments using an antibody against GFP corroborated that residual levels of SWI1 remain on chromatin in *rec8* mutants that expressed the SWI1-GFP reporter construct. This suggested that chromatin association of SWI1 also relies on other factors in addition to the REC8-containing cohesin (Supplementary Fig. 8a).

**SWI1 interacts with PDS5 family proteins**. A direct interaction of SWI1 with one of the cohesin components is a likely explanation for the observation that proper SWI1 localization is dependent on cohesin. To explore this possibility, we tested the interaction of SWI1 with all core cohesin subunits including SMC1, SMC3, REC8, and SCC3 by yeast two-hybrid assays. However, SWI1 did not interact with any of these proteins

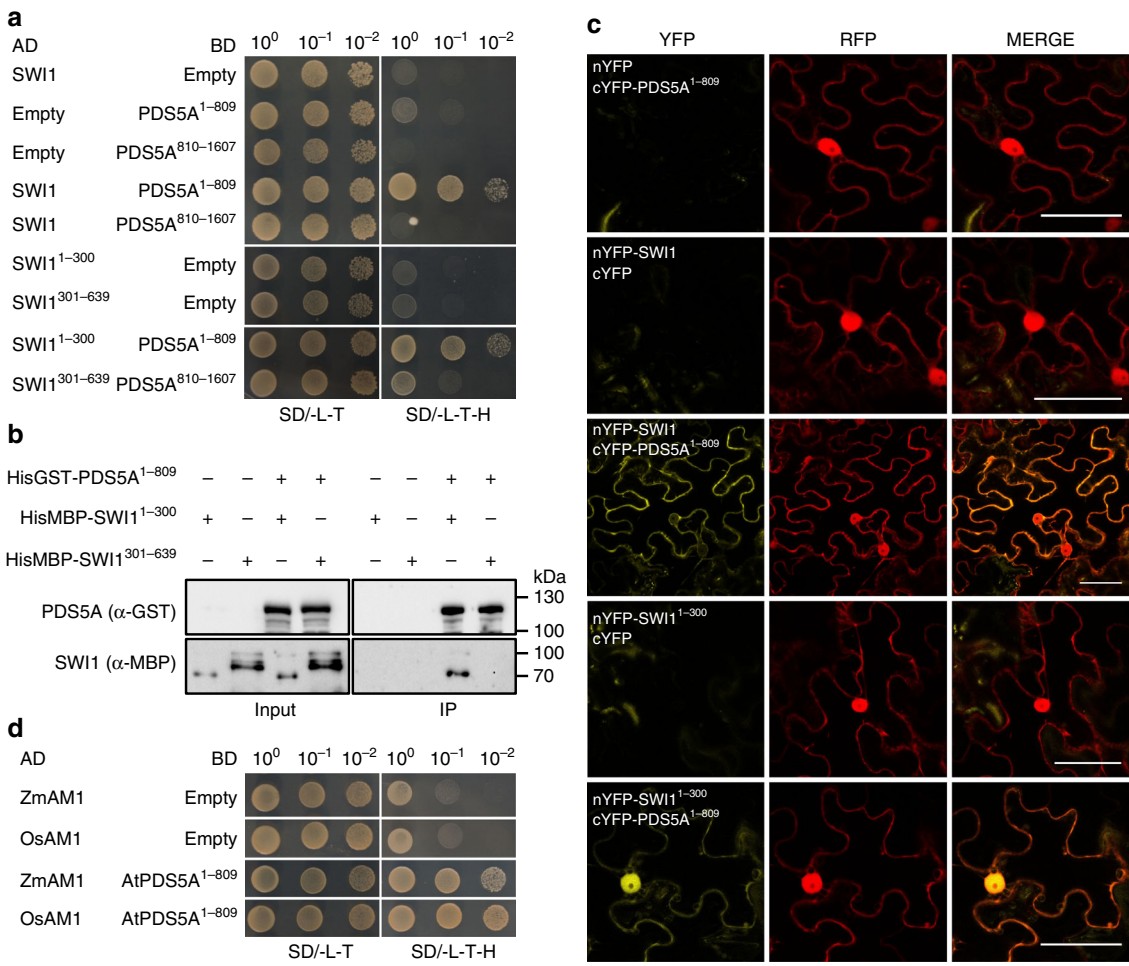

**Fig. 3** SWI1 interaction with cohesin components. **a** Yeast two-hybrid interaction assay of SWI1 with PDS5A. SWI1 and PDS5A were divided into an N-terminal part (SWI$^{1-300}$, PDS5A$^{1-809}$) and a C-terminal part (SWI$^{301-639}$, PDS5A$^{810-1607}$). Yeast cells harboring both the AD (activating domain) and BD (binding domain) were grown on synthetic medium supplied with dextrose (SD) in the absence of Leu and Trp (SD/ -L -T, left panel) and on SD medium in the absence of Leu, Trp, and His (SD/ -L -T -H, right panel). Yeast cells were incubated until $OD_{600} = 1$ and then diluted 10- and 100-fold for assays. **b** Co-immunoprecipitation assay of SWI1 with PDS5A. HisGST-PDS5A$^{1-809}$-bound or unoccupied agarose beads were incubated in the presence of HisMBP-SWI1$^{1-300}$ and HisMBP-SWI1$^{301-639}$. The pull-down fractions were analyzed by immunoblotting with anti-GST (upper panel) and anti-MBP (lower panel) antibodies. The source data of the uncropped immunoblots are provided in the Source Data file. **c** Interaction of SWI1 with PDS5A using bimolecular fluorescence complementation (BiFC) assays. YFP fluorescence indicates a successful complementation and hence interaction of the proteins tested. RFP is used as an indicator for the successful *Agrobacterium* infiltration. **d** Yeast two-hybrid interaction assay of SWI1 homologs in maize (ZmAM1) and rice (OsAM1) with *Arabidopsis* PDS5A (PDS5A)

(Supplementary Fig. 9a). We further investigated the interaction of SWI1 with the cohesin accessory proteins PDS5A, one of the five *PDS5* genes in *Arabidopsis*, and WAPL1, one of the two WAPL homologs. While we did not find an interaction of SWI1 with WAPL1, SWI1 strongly interacted with the N-terminus but not the C-terminus of PDS5A (Fig. 3a; Supplementary Fig. 9b). The interaction domain of SWI1 was then determined to reside in the N-terminal 300 amino acids as the C-terminal domain from amino acid 301-639 failed to bind to N-terminus of PDS5A (Fig. 3a). This interaction was confirmed by GST pull down assay with recombinant proteins purified from *E. coli*, and by bimolecular fluorescence complementation (BiFC) assay in tobacco leaves (Fig. 3b, c). Whether SWI1 also interacts with the other four PDS5 paralogs present in *Arabidopsis*, was next addressed in BiFC assay. While PDS5B and PDS5D only weakly bound to SWI1, an even stronger interaction of SWI1 with PDS5C and PD55E than with PDS5A was found, indicating that SWI1 has the potential to regulate all PDS5 proteins in *Arabidopsis*.

**SWI1 antagonizes WAPL**. PDS5 has been shown to form a complex with WAPL in several vertebrates and yeast[12,13,36,37]. Correspondingly, we found that *Arabidopsis* WAPL1 bound to the N- but not the C-terminus of PDS5A by yeast two-hybrid and BiFC assays (Supplementary Fig. 9b, c). Thus, WAPL1 and SWI1 interact, at least broadly, with the same region of PDS5. Sororin is known to bind to PDS5 and displace WAPL from the cohesin complex[18]. To assess whether SWI1 may act similarly as Sororin by dislodging WAPL from PDS5, we first compared the binding affinity of PDS5A with SWI1 and WAPL1 by using a ratiometric BiFC (rBiFC) system[38] that allows quantification of the interaction strength. The rBiFC assay revealed that the interaction between SWI1 and PDS5A is stronger than the interaction of WAPL1 with PDS5A (Fig. 4a, b). To further explore the relationship of these three proteins, we perform an in vitro competition experiment. To this end, we loaded recombinant WAPL1-PDS5A heterodimers co-purified from *E. coli* onto PDS5A-bound beads and incubated them with increasing concentrations of SWI1. With increasing concentrations of SWI1, more WAPL1

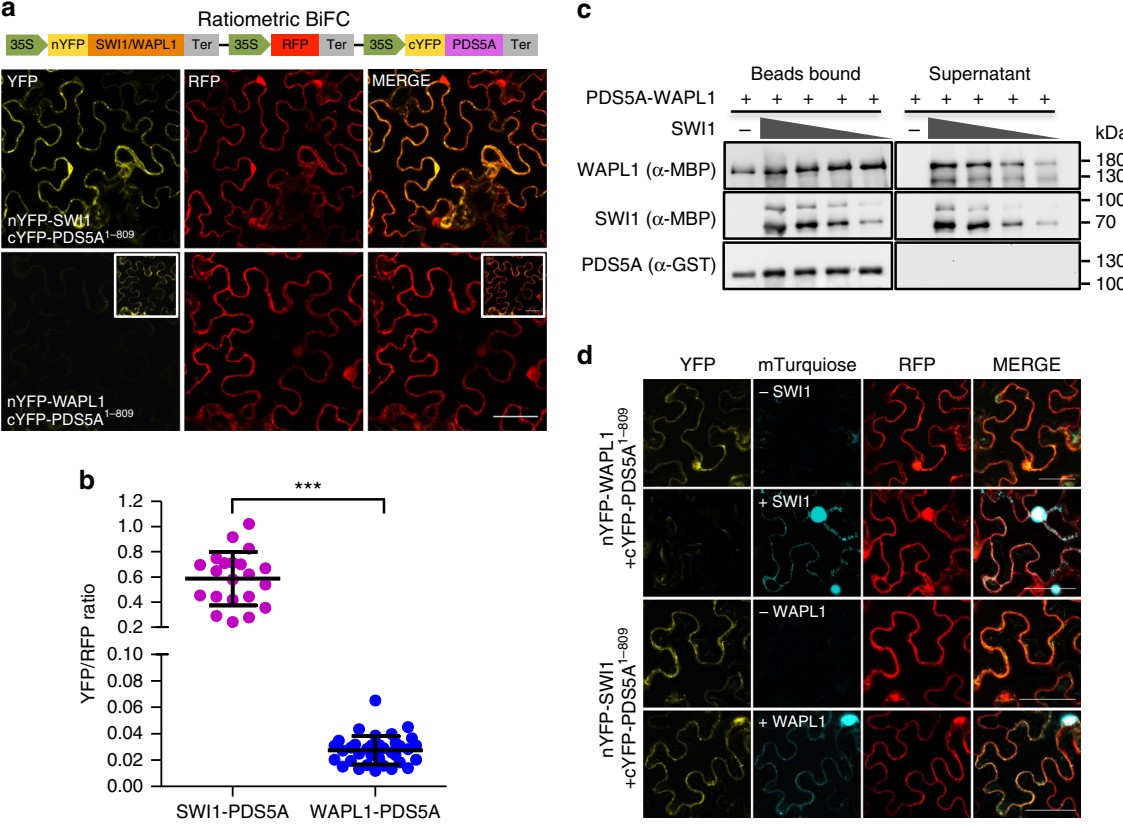

**Fig. 4** SWI1 dissociates WAPL1 from PDS5A. **a** Ratiometric BiFC (rBiFC) assays of PDS5A with SWI1 and WAPL1. The upper panel depicts the ratiometric gene expression cassette, and the below panels show representative images of the assay that were captured with the same settings at a confocal laser scanning microscope. The level of YFP fluorescence indicates the interaction strength with the RFP fluorescence used as a reference. The images in the white box represent the same pictures as the ones shown in the respective panel but taken with increased sensitivity revealing an interaction between WAPl1 and PDS5A. Bar: 50 μm. **b** Quantification of the rBiFC assay by calculating the ratio between YFP and RFP signal intensity shown in **a**. Asterisks indicate significant difference (Student's $t$-test, $P < 0.001$). Error bars represent standard deviations. **c** SWI1 causes the dissociation of WAPL from PDS5. Anti-GST beads were incubated with or without SWI1[1-300] in the presence or absence of PDS5A[1-809]-WAPL1 heterodimers. PDS5A[1-809] is His-GST tagged. WAPL1 and SWI1[1-300] are His-MBP tagged. Beads bound proteins were separated from the supernatant and analyzed by immunoblotting. Different amounts of SWI1[1-300] were used for the experiment. The empty beads control was shown in Supplementary Fig. 9e. The source data of the uncropped immunoblots are provided in the Source Data file. **d** Co-expression of SWI1-mTurquoise inhibits the interaction of WAPL1 with PDS5A in tobacco leaf cells while the presence of WAPL1-mTurquoise has no obvious impact on the interaction of SWI1 with PDS5A. Bar: 50 μm

protein could be released from PDS5A into the supernatant (Fig. 4c). Conversely, more SWI1 was bound to PDS5 with increasing concentrations of SWI1.

The displacement of WAPL from PDS5 by SWI1 was further confirmed by a competitive binding assay in tobacco leaf cells (Fig. 4d). While the simultaneous presence of WAPL1 tagged with mTuquiose did not affect the interaction of SWI1 with PDS5A, the co-expression of SWI1-mTurquiose resulted in a strong reduction of the BiFC signal from WAPL1-PDS5A interaction (Fig. 4d). Thus, despite any sequence similarities, SWI1 appears to act in a similar fashion as Sororin in animals.

Therefore, we speculated that the absence of WAPL should restore the presence of REC8 on chromatin in *swi1* mutants. To this end, we generated the triple mutant *swi1 wapl1 wapl2* containing in addition the *REC8-GFP* reporter. REC8 localization was then analyzed in male meiocytes at different meiotic stages of this triple mutant in comparison to the wildtype, *swi1* and *wapl1 wapl2* double mutants. In contrast to *swi1* mutants (Fig. 5a, b; Supplementary Fig. 7; Supplementary Movie 3), REC8 localization in *swi1 wapl1 wapl2* mutants was nearly identical to the pattern found in *wapl1 wapl2* double mutants, i.e., residing on chromosomes till metaphase I (Figs. 1a, b and 5c, d; Supplementary Movie 4). Note that due to the failure of chromosome axis

formation and of the aberrant migration of nucleolus in *swi1* mutants, the meiotic stages in *swi1* mutants were determined by the morphology of meiocytes in combination with the number of nuclei in tapetal cells[29,34]. The restoration of cohesion in the *swi1 wapl1 wapl2* and the resemblance to the *wapl1 wapl2* mutant phenotype was further confirmed by chromosome spread analysis (Fig. 5c). Since *swi1* mutants do not have an obvious growth defect and since we also could not detect SWI1 outside of meiocytes, we conclude that SWI1 specifically maintains cohesion in meiosis by antagonizing WAPL. We also found that the putative SWI1 homologs from maize and rice, *AMEIOTIC 1* (*AM1*), which likewise are required for meiotic progression and cohesion establishment[39,40], both interacted with *Arabidopsis* PDS5A in a yeast two-hybrid interaction assay (Fig. 3d). Thus, it is likely that the SWI1 function as a WAPL antagonist in meiosis is conserved in flowering plants and, given the presence of SWI1 homologs in moss, possibly in all land plants.

**SWI1 presence is controlled by Cdk-cyclin activity.** A crucial question is how WAPL is liberated from the inhibition by SWI1 in late prophase to mediate the release of cohesin (Fig. 1a–c). In vertebrate mitosis, this problem is solved by the phosphorylation

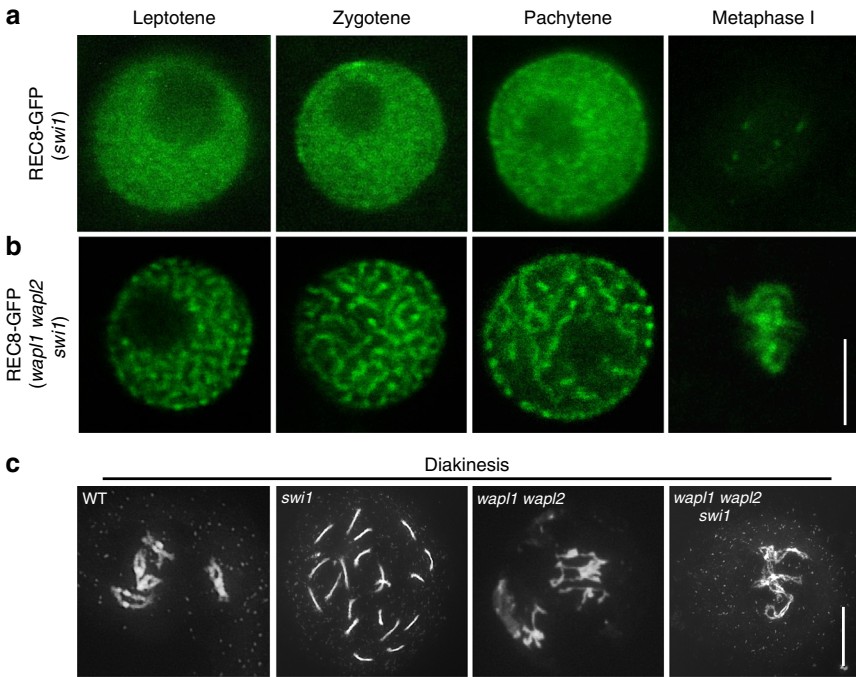

**Fig. 5** SWI1 is dispensable for the sister chromatid cohesion in the absence of WAPL. **a**, **b** Confocal laser scanning micrographs of REC8-GFP localization in male meiocytes in *swi1* (**a**) and in *swi1 wapl1 wapl2* (**b**). Bar: 5 μm. **c** Chromosome spreads of the wildtype (WT), *swi1*, *wapl1 wapl2* and *swi1 wapl1 wapl2* mutants in diakinesis. Bar: 10 μm

dependent release of Sororin from chromatin. Two kinases have been observed to participate in this regulation, Cyclin-dependent kinase 1 (Cdk1) and Aurora B[22,23]. We observed that SWI1 contains 13 consensus Cdk phosphorylation sites, 12 [S/T]P and 1 [S/T]Px[R/K] sites. We found that at least 7 of these sites can be phosphorylated in an in vitro kinase assay by CDKA;1, the *Arabidopsis* Cdk1/Cdk2 homolog, together with the meiotic cyclin SOLO DANCERS (SDS) (Fig. 6a; Supplementary Table 1).

To address whether the analogies between SWI1 and Sororin would extend to phospho-regulation, we introgressed the *SWI1-GFP* reporter, together with the *ASY3-RFP* reporter for staging, into weak loss-of-function alleles of *cdka;1* (*CDKA;1^{T161D}*)[41]. Similar to the wildtype, SWI1 is present on chromatin in *CDKA;1^{T161D}* plants until leptotene (Fig. 6b). However, the SWI1 signal does not decline as strongly in *CDKA;1^{T161D}* plants as in the complemented *swi1* mutants. Remarkably, SWI1 stayed associated with chromosomes even until pachytene (Fig. 6b). Similarly, SWI1-GFP was also prolonged present in meiocytes of *sds* mutants (Fig. 6c).

To test whether the phosphorylation of SWI1 is essential for its release from chromosomes at late prophase I, we generated dephospho mutant constructs. The localization pattern of SWI1 with four mutated CDK phosphorylation sites in the N-terminus of SWI1 (SWI1^{4A}-GFP), was indistinguishable from wildtype SWI1 protein (Fig. 6c). However, mutating all 13 or only the C-terminal nine phosphorylation sites in SWI1 (SWI1^{13A}-GFP and SWI1^{9A}-GFP), resulted in extended occupancy of SWI1 on chromosomes, reminiscent of the pattern found in *CDKA;1^{T161D}* and *sds* mutants (Fig. 6c; Supplementary Fig. 8b). Note that SWI1^{13A}-GFP and SWI1^{9A}-GFP seems to be functional since the cohesion defects in early prophase I were completely rescued in *swi1* mutants harboring either version (Supplementary Fig. 10d, e; for effects in later stages of meiosis, see below).

To complement this analysis, we also generated a phospho-mimic version of SWI1 in which the Serine or Threonine of all 13 CDK phosphorylation sites were mutated to the negatively charged amino acid Aspartate (SWI1^{13D}-GFP) and introduced

this construct into wild-type plants. SWI1^{13D}-GFP showed the same localization pattern as the wild-type version, indicating that the phosphomimic SWI1 version is recognized by its releasing factors (Fig. 7). Moreover, we did not find any reduction in fertility of these plants (Supplementary Fig. 11).

Taken together, these findings corroborated that mitosis in vertebrates and meiosis in plants (*Arabidopsis*) utilize a similar mechanism to control the presence of the WAPL inhibitors on chromatin through phosphorylation by CDK-cyclin complexes. However, the observation that SWI1 was not prematurely removed from chromatin by mimicking its phosphorylation indicates that phosphorylation is necessary but not sufficient for SWI1 removal hinting at a higher order coordination of SWI1 phosphorylation and the machinery involved in controlling its stability.

**Chromatin release of SWI1 is important for WAPL action.** Our above presented cytological and biochemical data suggested that the timely release of SWI1 is needed for WAPL to remove cohesin. To test this in vivo, we made use of the dephospho-mutant version of SWI1^{13A}-GFP that complemented the early defects of *swi1* mutants (Supplementary Fig. 10d, see above). Notably, *swi1* mutant harboring SWI1^{13A}-GFP were to a large degree infertile as seen by their short siliques and strongly reduced pollen viability (Supplementary Fig. 10a–c,i). Since these defects precluded discerning between a dominant effect as expected from interfering with WAPL versus a partial functionality of SWI1^{13A}-GFP, we switched to wild-type plants harboring the SWI1^{13A}-GFP construct (*SWI1^{13A}-GFP*/WT) for the following analysis. While the vegetative growth of these plants was not affected, they also suffered from a drastic fertility reduction in 51 out of 55 T1 transformants similar to *swi1* mutants expressing the *SWI1^{13A}-GFP* mutant version (Supplementary Fig. 10a–c, i), indicating that it is not the lack of a functional version that causes this phenotype.

Chromosome spread analysis showed that chromosome pairing and synapsis was not altered in *SWI1^{13A}-GFP*/WT

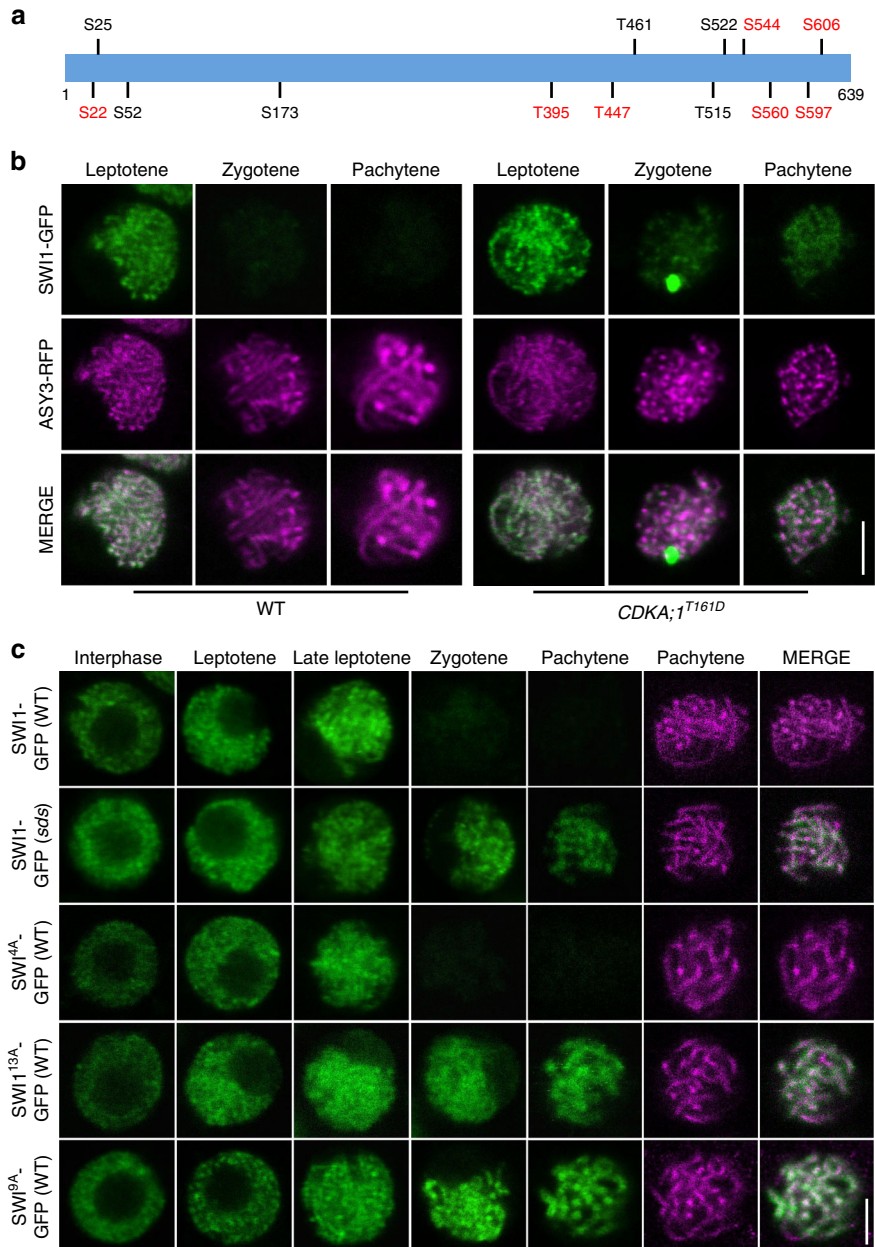

**Fig. 6** Phospho-control of SWI1 localization. **a** Schematic representation of SWI1 with the position of the 13 [S/T]P motifs. Phosphorylated sites identified by mass spectrometry are labeled in red. S serine, T threonine. **b** Confocal laser scanning micrographs of SWI1-GFP in comparison with ASY3-RFP as a meiosis staging marker in the wildtype (WT) and *CDKA;1^T161D* male meiocytes. **c** The expression of SWI1-GFP and the de-phospho mutants SWI1^4A-GFP, SWI1^9A-GFP and SWI1^13A-GFP were analyzed in interphase and prophase I of male meiocytes of *sds* mutants and wild-type plants (WT), respectively. ASY3-RFP localization is only shown for pachytene. Bar: 5 μm

($n = 88$) consistent with the restoration of these defects in *swi1* mutants by the same construct (Fig. 8ai). The first obvious defects were found at diakinesis. Whereas 5 clearly discernable bivalents are then present in the wildtype, chromosomes were entangled and clustered in *SWI1^13A-GFP*/WT (51 out of 101 meiocytes analyzed) (Fig. 8aii, xvi). Intertwined chromosomes of *SWI1^13A-GFP*/WT persisted until metaphase I (87 out of 190 meiocytes analyzed) (Fig. 8a iii, ix, x, xvii). After metaphase I, chromosome fragmentation was observed (Fig. 8a iv, xi, xviii). Entangled chromosomes were even found at metaphase II (30 out of 71 meiocytes analyzed) (Fig. 8a vi, xiii, xx). Finally, tetrads with an unequal amount of DNA and triads were frequently observed in *SWI1^13A-GFP*/WT (84 out of 156 meiocytes analyzed) (Fig. 8a vii, xiv, xxi; Supplementary Fig. 12). Taken together, *SWI1^13A-GFP*/

WT plants have an over cohesive phenotype which closely resembled the defects of the *wapl1 wapl2* mutants.

We therefore speculated that the prolonged retention of SWI1 might result in an extended abundance of cohesin on chromatin. To address this question, plants expressing a SWI1^13A version without a fluorescent tag were generated and combined with plants harboring the *REC8-GFP* reporter. Based on the time-resolved quantification of REC8-GFP signal in male meiocytes, we found that in comparison to wildtype, REC8-GFP signal showed a decreased speed of removal in *SWI1^13A* plants (1/2 removal time, $14.66 \pm 0.58$ h, $n = 3$ in *SWI1^13A* versus $11.33 \pm 1.15$ h, $n = 3$ in wildtype) (Fig. 8b; Supplementary Movies 5 and 6). At metaphase I, instead of ~10% ($n = 3$) REC8-GFP signal retained in the wildtype, twice the signal, i.e., ~20% ($n = 3$) was

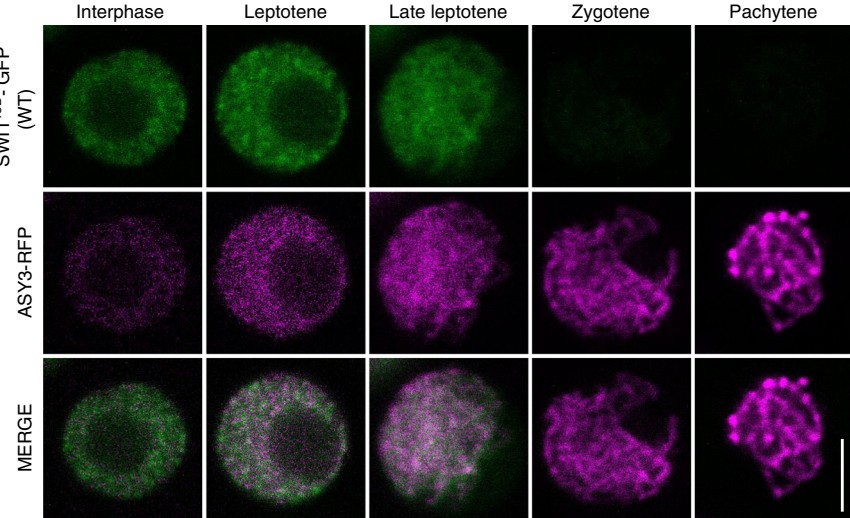

**Fig. 7** Localization of the phosphomimic version of SWI1. The localization of the phosphomimic version SWI1[13D]-GFP is indistinguishable from the wild-type SWI1-GFP version (compare with Fig. 2a). ASY3-RFP is used for staging. Bar: 5 µm

observed in *SWI1[13A]* plants (Student's *t*-test $P < 0.0001$) partially resembling the retention of REC8-GFP in *wapl1 wapl2* mutants. However, it has to be noted that the level of REC8-GFP withholding in *wapl1 wapl2* is higher than in *SWI1[13A]* plants (~55% versus ~20%) (Figs. 1a and 8b; Supplementary Movies 2, 5, and 6). The reason for this is not clear and we cannot exclude a slightly altered biochemical property of *SWI1[13A]* due to the substitution of 13 amino acids possibly resulting in a less efficient inhibition of WAPL. Consistent with such a scenario is the observation that the eviction of REC8 starts apparently earlier in *SWI1[13A]* versus the wildtype (Fig. 8b). In any case, our data strongly suggest that a vast (more than 90%) removal of REC8 is crucial for meiosis and even an increase from 10 to 20% is sufficient to cause an over cohesive effect underlining the importance of the WAPL-PDS5-SWI1 regulatory triangle.

**SWI1 abundance is controlled by the APC/C.** Our results show that the release of SWI1 from chromosomes is regulated by CDKA;1-mediated phosphorylation. However, the degradation pathway for SWI1 is still obscure. An analysis of SWI1 by the GPS-ARM algorithm[42] revealed five putative destruction boxes (D-box) in the C-terminus of SWI1, including 2 canonical and 3 less conserved D-boxes, hinting at a potential regulation of SWI1 by the APC/C (Supplementary Fig. 13a).

To address the functional relevance of the predicated D-boxes, we first mutated the two conserved D-boxes at position 306-309 and 559-562 (RXXL to AXXA) and generated plants expressing this *SWI1* mutant version (*SWI1-Δ2D-box-GFP*). Since plants harboring *SWI1-Δ2D-box-GFP* had no any obviously altered SWI1 protein expression and localization pattern (Supplementary Fig. 13b), we mutated all 5 D-boxes (*SWI1-Δ5D-box-GFP*). Plant expressing *SWI1-Δ5D-box-GFP* showed an extremely prolonged abundance of SWI1 that only disappeared in tetrads, suggesting that SWI1 is targeted by the APC/C for degradation from zygotene onwards (Fig. 9a; Supplementary Fig. 10f–i). We also observed reduced fertility of *SWI1-Δ5D-box-GFP* expressing plants consistent with the prolonged presence of SWI1 on chromatin. However, the reduction in fertility was less severe in plants expressing *SWI1-Δ5D-box-GFP* than in *wapl1 wapl2* mutants or in *SWI1[13A]-GFP* expressing plants (Supplementary Fig. 10f–i). Again, we cannot exclude a compromised function of SWI1-Δ5D-box-GFP due to the many point mutations introduced and, consistent with an affected functionality, we also

observed that SWI1-Δ5D-box-GFP had a slightly less pronounced chromosome association than the non-mutated SWI1-GFP version (compare Fig. 9a with Fig. 2a).

To hence seek further evidence for a possible proteolytic control of SWI1, we performed a cell free degradation assay by incubating protein extracts from flower buds with the purified C-terminal half of the SWI1 protein (HisGST-SWI1[301-639]). We found that SWI1[301-639] degradation started in mock-treated samples after 15 min of incubation time and the majority of the protein (80%) was not detectable any longer by 90 min. In contrast, SWI1[301–639] disappeared at a much slower rate in samples treated with the proteasome inhibitor MG132 and after 90 min, more than 50% of the protein was still present (Fig. 9b i, ii, c).

Since we found that phosphorylation is required for the release of SWI1 from chromatin, we next compared the degradation kinetics of wild-type SWI1[301-639] with the mutated SWI1[301-639/9A] version. Indeed, the non-phosphorylatable version SWI1[301-639/9A] was stabilized in comparison to the phosphorylatable version and showed similar turnover kinetics as MG132-treated extracts (Fig. 9b iii, c). To further assess whether the degradation of SWI1 is mediated through the phosphorylation of SWI1 by CDKs, we treated the protein extracts with Roscovitine, a potent CDK inhibitor[43]. In comparison to the mock-treated sample, SWI1[301-639] was also stabilized under Roscovitine treatment, substantiating that CDK-dependent phosphorylation marks SWI1 for 26S proteasome-dependent degradation which relies on D-boxes and thus, is likely mediated by the APC/C.

**Discussion**

The precise establishment, maintenance, and removal of sister chromatid cohesion is essential for faithful chromosome segregation in both mitosis and meiosis. In contrast to the well-described mechanisms of cohesion regulation in mitosis[17,18,20], much less is known about the control of cohesion in meiosis. Our study in *Arabidopsis* provides evidence that the prophase pathway of cohesion regulation exists in meiosis including the inhibition of the cohesin remodeler WAPL by a new type of inhibitor represented by the previously identified protein SWI1 that functions and is regulated in an amazingly similar fashion as Sororin in animals. Given that both animals and plants have WAPL homologs and that the lineage that led to plants and to animals were very early separated in eukaryotic evolution, much earlier than the separation of the predecessors of animals and yeast, it is

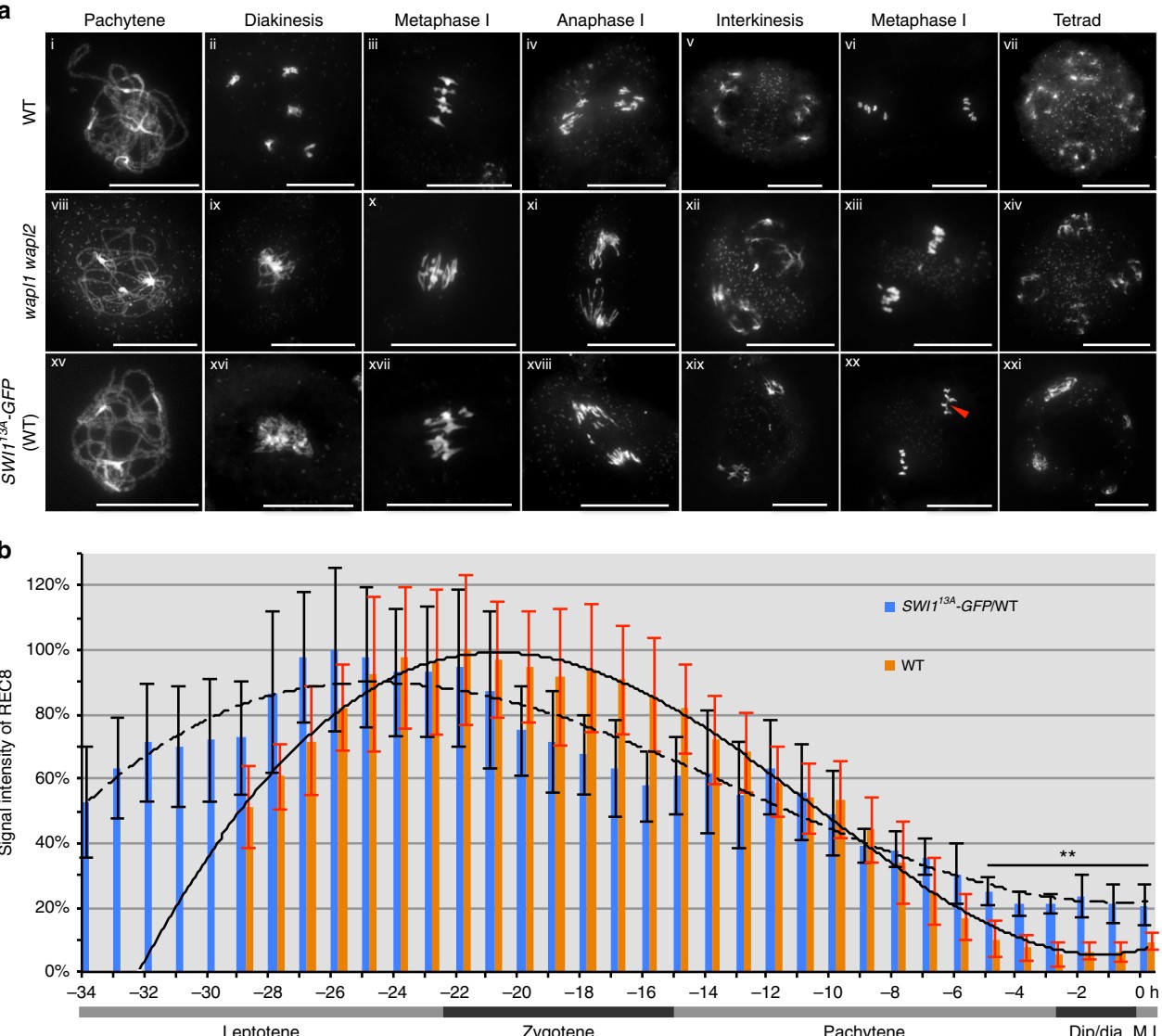

**Fig. 8** De-phosphomimic SWI1^13A-GFP interferes with the cohesin removal. **a** Chromosome spreads of male meiocytes of the wildtype (WT), *wapl1 wapl2* mutants, and *SWI1^13A-GFP* in wild-type plants (*SWI1^13A-GFP*/WT). Bar: 20 μm. Red arrowhead in xx highlights intertwined chromosomes. **b** Quantification of cohesin during meiosis I in male meiocytes of the wildtype (WT) and *SWI1^13A-GFP*/WT based on the analysis of a REC8-GFP reporter. The graph represents the relative intensity of the REC8-GFP signal; error bar represents standard deviation of at least 10 meiocytes analyzed. Asterisks represent significant difference (Student's *t*-test, *P* < 0.01). Dip/dia diplotene/diakinesis, M I metaphase I. A solid polynomial trendline is shown for the wildtype and a dashed line for *SWI1^13A-GFP*/WT (correlation coefficient $R^2 = 0.993$ for the wildtype and 0.942 for *SWI1^13A-GFP*/WT). The source data of this graph are provided in the Source Data file

likely that a basic prophase pathway of cohesin removal is very ancient and was probably present in the last common ancestor of animals and plants.

However, the repression of WAPL appears to have evolved independently in animals and plants and hence is likely younger in evolutionary terms. Remarkably, the two independent WAPL regulators, Sororin and SWI1, target the same cohesin subunit, i.e., PDS5, and are themselves controlled by a similar mechanisms, i.e., Cdk phosphorylation. Our finding that SWI1 from *Arabidopsis* can also bind to PDS5 from maize and rice indicates that the function of SWI1 as a WAPL antagonist is conserved in flowering plants. Moreover, the presence of *SWI1* homologs in moss gives rise to the hypothesis that *SWI1* appeared very early in the plant lineage.

Based on our results, we propose the following model of how SWI1 prevents the premature removal of sister chromatid cohesion in *Arabidopsis* (Fig. 10): SWI1 starts to be expressed and is

recruited onto chromosomes by interacting with PDS5 proteins during very early meiosis, likely already during the premeiotic S phase. The recruitment of SWI1 is dependent, at least partially, on the formation of the cohesin complex (Fig. 2c). After entry into meiosis, cohesin needs to be maintained until late prophase to facilitate multiple processes of meiosis, e.g., double-strand break (DSB) repair, chromosome pairing, and homologous recombination[44,45]. The maintenance of cohesin is achieved by SWI1 that has a higher affinity towards PDS5 than WAPL, thereby displacing WAPL from PDS5, consistent with the dispersed localization of WAPL at early stages in prophase I (Fig. 1d). Given the stronger interaction strength between SWI1 and PDS5 versus WAPL and PDS5, it seems likely that both complexes do not co-exist or that at least the SWI1-PDS5 complex is much more prominent than a WAPL-PDS5 complex if all three proteins are equally present.

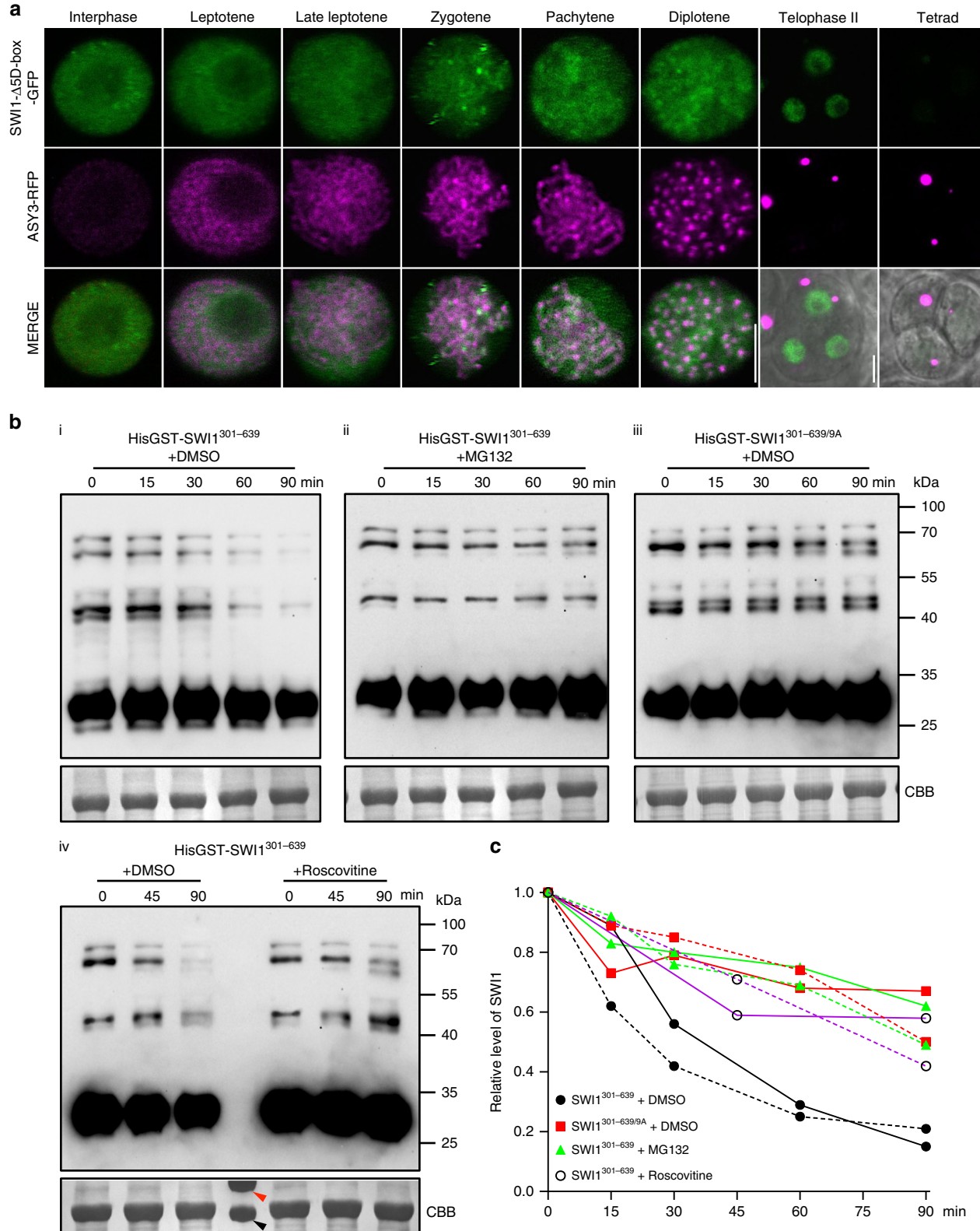

**Fig. 9** APC/C-mediated degradation of SWI1. **a** Deletion of the five predicted destruction boxes (D-box) in SWI1 (SWI1-Δ5D-box-GFP) results in prolonged occupancy along chromatin in comparison with SWI1-GFP (Fig. 2a). ASY3-RFP is shown for staging. Bar: 5 μm. **b** Cell free degradation assay of HisGST-SWI$^{301-639}$ and HisGST-SWI$^{310-639}$ in the presence of DMSO (solvent control), 50 μm MG132, or 5 μm Roscovitine. The source data of the uncropped immunoblots are provided in the Source Data file. **c** Relative protein level of SWI1 according to **b**. The intensity of all bands between 100 and 40 kDa was measured and plotted on the graph. The solid lines represent the relative protein level of SWI1 shown in **b** and the dashed lines depict the results of the second biological replicate. The large subunit of Rubisco stained by CBB verifies the equal loading of the samples. The red and black arrowheads indicate the protein marker at 70 and 55 kDa, respectively

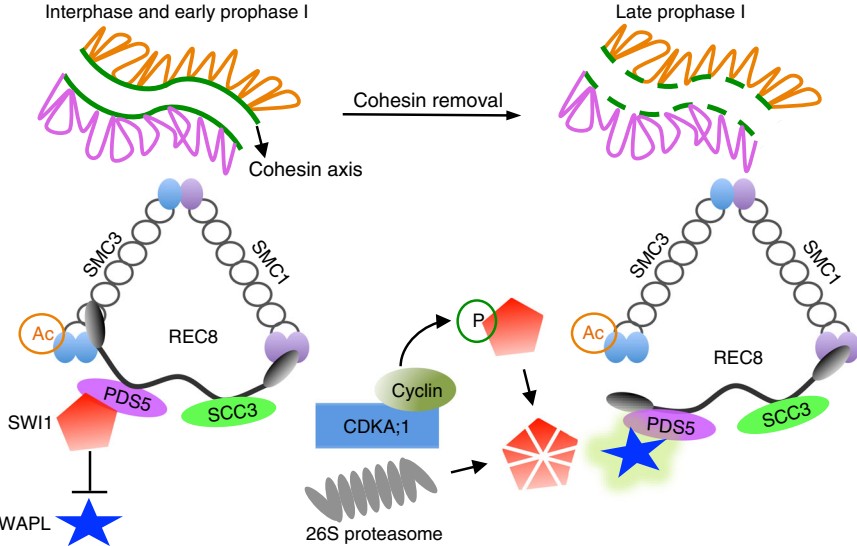

**Fig. 10** Model for the role of SWI1 in the regulation of cohesin. During interphase and early prophase I, SWI1 is highly expressed and is recruited to chromatin through interacting with PDS5 family proteins, which in turn inhibits the action of WAPL by dislodging WAPL from PDS5. In late prophase I, SWI1 is phosphorylated by CDKA;1. This phosphorylation of SWI1 then promotes its release from chromatin by facilitating the ubiquitination by APC/C, and subsequent degradation by the 26S proteasome. The release of SWI1 allows the binding of WAPL to PDS5 resulting in the release of cohesin from chromatin

While a protein sequence-based alignment of the first 300 amino acids of the *Arabidopsis* SWI1 with its otologs in other plant species including Brachypodium, bean, maize, sorghum, rapeseed, and rice revealed three conserved domains, no clear motif is emerging and further work will be required to address whether one of these domains or a specific combination mediates the interaction with PDS5. Including Sororin in this alignment did also not pinpoint to a PDS5 binding domain making it likely that the interaction between WAPL inhibitors and PDS5 is complex.

SWI1 is released from chromatin by CDKA;1-dependent phosphorylation and subjected to degradation, likely in an APC/C-mediated manner (Fig. 10). However, CDKA;1 phosphorylation of SWI1 does not appears to be sufficient for degradation and possibly, the degradation machinery needs to be activated as well, perhaps depending on CDKA;1 phosphorylation as well. The removal of SWI1 allows the interaction of WAPL with PDS5 as indicated by the tight chromosome association of WAPL at late prophase (Fig. 1d; Supplementary Fig. 1c), thereby activating the prophase pathway of cohesin removal.

The retained signal of SWI1-GFP in *rec8* mutants suggests that SWI1 might also localize to other cohesin complexes that do not contain REC8. Unlike most other organisms that have only one mitotic *RAD21* gene, three paralogs of the kleisin subunit, *RAD21.1/SYN2*, *RAD21.2/SYN3*, and *RAD21.3/SYN4*, have been identified in *Arabidopsis* next to the meiotic paralog REC8/SYN1. Similarly rice and other plants also have several *SCC1/RAD21* genes[1]. Among the *Arabidopsis* RAD21 genes, especially RAD21.2 has been found to play an important role in reproduction, i.e., meiosis and subsequent gametogenesis, next to a function in vegetative growth since knockdown of RAD21.2 in meiocytes impaired chromosome synapsis and SC formation[46,47]. This together with the observation that sister chromatids are still bound at centromeres in the absence of *REC8* until metaphase I indicates that at least two different kleisins contribute to sister chromatid cohesion. However, RAD21.2 was unexpectedly detected to be predominantly present in the nucleolus of meiocytes and not along chromatin[46]. Thus, the function of this putative kleisin is still obscure and it is also not clear whether SWI1 can regulate different types cohesin complexes in meiosis. Conversely, *wapl* and *swi1* mutants do not show any mitotic

defects raising the question whether there is a prophase pathway in mitosis in *Arabidopsis*.

Although the premature removal of REC8 and with that the REC8-dependent cohesion loss are suppressed by the absence of *WAPL1* and *WAPL2*, *swi1 wapl1 wapl2* plants are still completely sterile similarly to the *swi1* single mutants and much more affected than *wapl1 wapl2* double mutants (Supplementary Fig. 14). This implies that SWI1 might have further roles in meiosis and/or might be involved in other biological processes after meiosis. Indeed, in addition to the cohesion defects, *swi1* mutants are also compromised in the specification of meiocytes resulting in the formation of multiple rather than a single female meiocyte[48,49]. These defects are specific to *swi1* in *Arabidopsis* and have not been reported for *am1* mutants in maize and rice[39,40]. However, AM1 also appears to regulate the entry into meiosis and very early meiotic events.

Interestingly, we found that the number of ovules with a single female meiocyte is significantly higher in the *swi1 wapl1 wapl2* mutants (14.3%, $n = 126$) than in *swi1* mutants (3.9%, $n = 128$) (Chi-squared test, $P = 0.00395$, Supplementary Fig. 15). Whether the loss of cohesin contributes to the formation of multiple meiocytes is not clear as yet. The germline in plants has to be established post-embryonically and there is accumulating evidence that the specification of meiocytes also involves a substantial reprogramming of chromatin possibly contributing to the repression of mitotic regulators and stem cell genes[50–52]. In this context it is interesting to note that the pattern of histone modifications is altered in *swi1* mutants[53]. Our finding that SWI1 binds to PDS5 opens a new perspective here given that PDS5 plays a broad role in regulating plant growth and development[54]. Thus, it is tempting to speculate that PDS5 is also involved in meiocyte specification in *Arabidopsis*. Further work is required to explore the role of SWI1 as a regulator of PDS5 containing complexes.

## Methods

**Plant material and growth conditions**. The *Arabidopsis thaliana* accession Columbia (Col-0) was used as wild-type reference throughout this study. The T-DNA insertion lines SAIL_654_C06 (*swi1-4*), SAIL_423H01 (*asy3-1*)[55], SALK_146172 (*spo11-1-3*)[56], SAIL_807_B08 (*rec8*) and SALK_046272 (*asy1-4*)[57] were obtained from the collection of T-DNA mutants at the Salk Institute Genomic Analysis Laboratory (SIGnAL, http://signal.salk.edu/cgi-bin/tdnaexpress) and GABI_206H06 (*swi1-3*)[58]

was obtained from GABI-Kat T-DNA mutant collection (http://www.GABI-Kat.de) via NASC (http://arabidopsis.info/). The mutant *swi1-2* has a premature stop codon induced by EMS (ethyl methanesulfonate) and was kindly provided by Raphaël Mercier from INRA Centre de Versailles-Grignon. If not mentioned otherwise, *swi1* denotes *swi1-3*. All plants were grown in growth chambers with a 16 h light/21 °C and 8 h/18 °C dark photoperiod and 60% humidity.

**Plasmid construction and plant transformation**. To create the *SWI1* reporters, the genomic sequence of *SWI1* was amplified by PCR and inserted into *pDONR221* vector by gateway BP reactions (Supplemental Table 2). The *SmaI* restriction site was then introduced in front of the stop codon by PCR. All constructs were then linearized by *SmaI* restriction and ligated to GFP or RFP fragments, followed by gateway LR reactions with the destination vector *pGWB501*[59]. WAPL1-GFP and ASY3-RFP reporters were created by using same strategy as described above. For the ZYP1b-GFP reporter, the *AscI* restriction site was inserted into *pDONR221-ZYP1b* between the 464-465aa of ZYP1b by PCR. Following the linearization by *AscI*, the mEGFP fragment was inserted into *pDONR221-ZYP1b*. The resulting ZYP1b-GFP expressing cassette was integrated into the destination vector *pGWB501* by a gateway LR reaction. Primers used for the creation of these constructs are shown in Supplementary Table 2. All constructs were transformed into *Arabidopsis thaliana* plants by floral dipping.

**Yeast two-hybrid assay**. The coding sequences of the respective genes were amplified by PCR from cDNA with primers flanking the *attB* recombination sites and subcloned into *pDONR223* vector by gateway BP reactions (Supplementary Table 2). The resulting constructs were integrated into the *pGADT7-GW* or *pGBKT7-GW* vectors by gateway LR reactions. Yeast two-hybrid assays were performed according to the Matchmaker Gold Yeast two-hybrid system manual (Clontech). Different combinations of constructs were co-transformed into yeast strain AH109 using the polyethylene glycol/lithium acetate method as described in the manual of Clontech. Yeast cells harboring the relevant constructs were grown on the SD/-Leu -Trp and SD/-Leu -Trp -His plates to test for protein-protein interactions.

**Protein expression and purification**. To generate HisMBP-SWI1[1-300], HisMBP-SWI1[301-639], HisGST-PDS5A[1-809] and HisMBP-WAPL1 constructs, the respective coding sequences were amplified by PCR and subcloned into *pDONR223* vector by gateway BP reactions (Table S2). The resulting constructs were integrated by gateway LR reactions into *pHMGWA* or *pHGGWA* vectors for the His MBP- or HisGST- tagged fusions, respectively. For the heterologous expression, the constructs were transformed into the *E. coli* BL21 (DE3)pLysS cells and grown at 37 °C in the presence of 100 mg/l ampicillin until the $OD_{600}$ of 0.6. Protein expression was induced by adding IPTG to a final concentration of 0.3 mM, and the cells were incubated at 37 °C for 3 h (HisMBP-SWI1[301-639]) or 18 °C overnight (HisMBP-SWI1[1-300], HisGST-PDS5A[1-809] and HisMBP-WAPL1). All proteins were purified under native conditions by using Ni-NTA sepharose (QIAGEN) according to the manual.

For the purification of PDS5A[1-809]-WAPL1 heterodimers, the ampicillin resistance gene of WAPL1-pHMGWA was first replaced by the kanamycin resistance gene and co-transformed into BL21 (DE3)pLysS cells containing the vector PDS5A[1-809]-pHGGWA. The cells harboring both constructs were grown at 37 °C in the presence of 100 mg/l ampicillin and 50 mg/l kanamycin until the $OD_{600}$ to 0.6 and induced with 0.3 mM IPTG at 18 °C for overnight. Cells were harvested and PDS5A[1-809] and WAPL1 heterodimers were purified using GST agarose beads (Novogene). Coomassie Brilliant Blue (CBB) stained gels of all purified proteins used in this study are shown in Supplementary Fig. 15. The protease inhibitor cocktail (Roche) was always used in the purification procedures.

**In vitro protein binding affinity assay**. For the GST pull-down assay, 1 µg of HisGST-PDS5A[1-809], HisMBP-SWI1[1-300] and HisMBP-SWI1[301-639] protein, purified as described above, were added to the pull-down buffer system containing 25 mM Tris-HCl, pH 7.5, 100 mM NaCl, 10% glycerol, and 20 µl GST agarose beads (Chromotek) as indicated in Fig. 3c. After incubation for 1 h at 4 °C, the GST beads were rinsed 5 times by the washing buffer containing 25 mM Tris-HCl, pH 7.5, 200 mM NaCl, 10% glycerol and 0.1% Triton X-100. The beads-bound proteins were eluted by boiling in an equal volume of 1X SDS protein loading buffer and subjected to immunoblotting.

For the WAPL removal assay, 50 ng/µl HisGST-PDS5A[1-809]-HisMBP-WAPL1 heterodimers were bound to anti-GST agarose beads and incubated with different amounts of HisMBP-SWI1[1-300] (40, 80, 120 or 200 ng/µl) in a buffer containing 25 mM Tris-HCl, pH 7.5, 100 mM NaCl, 10 mM MgCl₂, 2 mM DTT, 10% glycerol and 0.1% Triton X-100 for 1 h at 4 °C. After incubation, the proteins in supernatant and from the beads-bound fraction were separated and subjected to immunoblot analysis. The GST (sc-138, 1:1000 dilution), MBP (E8032S, 1:10,000 dilution), and anti-mouse IgG secondary (A9044, 1:10,000 dilution) antibodies used were purchased from Santa Cruz Biotechnology, New England Biolabs, and Sigma, respectively.

**Chromosome spread analysis**. Chromosome spreads were performed as described previously[60]. In brief, fresh flower buds were fixed in 75% ethanol and 25% acetic acid for 48 h at 4 °C, followed by two washing steps with 75% ethanol and stored in 75% ethanol at 4 °C. For chromosome spreading, flower buds were digested in an enzyme solution (10 mM citrate buffer containing 1.5% cellulose, 1.5% pectolyase, and 1.5% cytohelicase) for 3 h at 37 °C and then transferred onto a glass slide, followed by dispersing with a bended needle. Spreading was performed on a 46 °C hotplate by adding an ~10 µl drop of 45% acetic acid. The slide was then rinsed by ice-cold ethanol/acetic acid (3:1) and mounted in Vectashield with DAPI (Vector Laboratories) to observe the DNA.

**In vitro kinase assay**. CDKA;1-SDS complexes were expressed and purified as described in Harashima and Schnittger[61] with slight modification using Strep-Tactin agarose (iba) instead of Ni-NTA agarose for the purification. Briefly, the CDKA;1-SDS complexes were first purified by Strep-Tactin agarose (iba), followed by desalting with PD MiniTrap G-25 (GE Healthcare). The quality of purified kinase complexes was checked by CBB staining and immunoblotting (Supplementary Fig. 15). Kinase assays were performed by incubating the kinase complexes with HisMBP-SWI1[1-300] or HisMBP-SWI1[1-639] in the reaction buffer containing 50 mM Tris-HCl, pH 7.5, 10 mM MgCl₂, 0.5 mM ATP and 5 mM DTT for 90 min. The CBB stained gel after kinase reaction is shown in Fig. S9.

**Cell-free degradation assay**. Wild-type *Arabidopsis* flower buds were freshly harvested and immediately grounded into powder in liquid nitrogen. Subsequently, total soluble proteins were extracted in the degradation buffer containing containing 25 mM Tris-HCl (pH 7.5), 10 mM NaCl, 10 mM MgCl₂, 4 mM PMSF, 5 mM DTT, and 10 mM ATP as previously described[62]. The supernatant was harvested after two 10-min centrifugations at 16,000 × *g* at 4 °C and protein concentration was measured using the Bio-Rad protein assay. Two hundred nanograms of recombinant proteins (HisGST-SWI1[301-639] or HisGST-SWI1[301-639/9A]) were mixed with 150 µl protein extracts (600 µg in total) containing either DMSO, 50 µm MG132, or 5 µm Roscovitine. The reactions were incubated at 22 °C and protein samples were collected at the indicated intervals followed by western blot analysis.

**Bimolecular fluorescence complementation assay**. The coding sequences of *SWI1*, *PDS5* paralogs and *WAPl1* were amplified from cDNA and subcloned into *pDONR221-P3P2* or *pDONR221-P1P4*. The resulting constructs were subsequently integrated into the *pBiFC-2in1-NN* destination vector using a gateway LR reaction[38]. All genes were under the control of the *35S* promoter. The relevant proteins were transiently expressed in tobacco leaves by *Agrobacterium* infiltration at a concentration of OD600. The fluorescence of YFP was imaged 2 days after infiltration using a Leica SP8 laser-scanning confocal microscope. For the competitive interaction BiFC assay in tobacco, *SWI1-mTurquoise2* and *WAPL1-mTurquoise2*, both driven by *35S* promoter, were generated by integrated their coding sequences into *pGWB502* vector. The resulting constructs were then brought into *Agrobacterium*. Co-infiltration was performed by mixing the *Agrobacterium* of *SWI1-mTurquoise2* and *WAPL1-mTurquoise2* with the *pBiFC-2in1-NN* harboring relevant combinations. YFP fluorescence was imaged 2 days after infiltration using a Leica SP8 laser-scanning confocal microscope with the same setup.

**Immunolocalization**. For immunolocalization analyses, freshly harvested young flower buds were sorted by different size and the intact anthers were macerated in 10 µl enzyme mix (0.4% cytohelicase with 1% polyvinylpyrrolidone) for 5 min in a moisture chamber at 37 °C followed by a squashing. Subsequently, 10 µl enzyme mix was added onto the Poly-Prep slides (Sigma) that were incubated for further 7 min in a moisture chamber. Afterwards, a fine smashing of the anthers was performed in 20 µl 1% Lipsol for 2 min. Cell fixation was then performed by incubating 35 µl 4% (w/v) paraformaldehyde for 2–3 h until dry. After three times washing with PBST buffer (PBS with 1% Triton X-100), slides were then blocked in PBS buffer with 1% BSA (PBSA) for 30 min at 37 °C in a moisture chamber followed by an incubation with anti-GFP (1:100 in PBSA) antibody at 4 °C for 72 h (Takara 632381/JL-8)). After three times of washing (10 min each) in PBST, fluorescein-conjugated horse anti-mouse antibody (FI-2000, Vector Laboratories) were added onto the slides (1:500 dilution) followed by 2 h incubation at 37 °C in a moisture chamber. After three times washing in PBST, the chromosomes were counterstained with anti-fade DAPI solution (Vector Laboratories). The images were captured using the Leica SP8 laser scanning microscopy.

**Sample preparation and LC-MS/MS data acquisition**. The protein mixtures were reduced with dithiothreitol, alkylated with chloroacetamide, and digested with trypsin. These digested samples were desalted using StageTips with C18 Empore disk membranes (3 M)[63], dried in a vacuum evaporator, and dissolved in 2% ACN, 0.1% TFA. Samples were analyzed using an EASY-nLC 1200 (Thermo Fisher) coupled to a Q Exactive Plus mass spectrometer (Thermo Fisher). Peptides were separated on 16 cm frit-less silica emitters (New Objective, 0.75 µm inner diameter), packed in-house with reversed-phase ReproSil-Pur C18 AQ 1.9 µm resin (Dr. Maisch GmbH). Peptides were loaded on the column and eluted for 50 min using a segmented linear gradient of

5% to 95% solvent B at a flow rate of 300 nl/min (0 min: 5%B; 0–5 min ->5%B; 5–25 min ->20%B; 25–35 min ->35%B; 35–40 min ->95%B; 40–50 min ->95%B; solvent A 0% ACN, 0.1% FA; solvent B 80% ACN, 0.1%FA). Mass spectra were acquired in data-dependent acquisition mode with a TOP15 method. MS spectra were acquired in the Orbitrap analyzer with a mass range of 300–1500 $m/z$ at a resolution of 70,000 FWHM and a target value of $3 \times 10^6$ ions. Precursors were selected with an isolation window of 1.3 $m/z$. HCD fragmentation was performed at a normalized collision energy of 25. MS/MS spectra were acquired with a target value of $5 \times 10^5$ ions at a resolution of 17,500 FWHM, a maximum injection time of 120 ms and a fixed first mass of $m/z$ 100. Peptides with a charge of 1, >6, or with unassigned charge state were excluded from fragmentation for MS$^2$; dynamic exclusion for 20 s prevented repeated selection of precursors.

**MS data analysis**. Raw data were processed individually using MaxQuant software (version 1.5.7.4, http://www.maxquant.org/)[64]. MS/MS spectra were searched by the Andromeda search engine against a database containing the respective proteins used for the in vitro reaction and a background *E. coli* database (*E. coli* (K12) database, UniProt). Sequences of 244 common contaminant proteins and decoy sequences were added during the search. Trypsin specificity was required and a maximum of two missed cleavages allowed. Minimal peptide length was set to seven amino acids. Carbamidomethylation of cysteine residues was set as fixed, phosphorylation of serine, threonine and tyrosine, oxidation of methionine and protein N-terminal acetylation as variable modifications. Peptide-spectrum-matches and proteins were retained if they were below a false discovery rate of 1%.

**Confocal microscopy and sample preparation**. For protein localization analyses, young anthers or ovules were dissected and imaged using an Leica TCS SP8 inverted confocal microscope. For tracing the dynamics of cohesin in *swi1 wapl1 wapl2* mutants, live cell imaging was performed as described by Prusicki et al.[29]. Briefly, one single fresh flower bud was detached from the flower and dissected with two anthers exposed. Subsequently, the isolated bud including the pedicel and a short part of the floral stem was embedded into the Arabidopsis Apex Culture Medium (ACM) and then covered by one drop of 2% agarose. The sample was then subjected to constant image capture with 15 min intervals by using an upright Zeiss LSM880 confocal microscope with Airyscan.

For analyzing the dynamics of cohesin during meiosis, live cell imaging was performed with the anthers of the wildtype and *wapl1 wapl2* mutant plants harboring a functional REC8-GFP reporter. To quantify the dynamics of the REC8-GFP signal, the metaphase I was denoted as 0 h and the REC8-GFP signal from at least 20 meiocytes was calculated for every one hour prior to metaphase I by using the image processing software Fiji[65]. Representative movies for the wildtype and *wapl1 wapl2* mutants are shown in the Supplementary Movie 1 and S2, respectively.

**Pollen viability assay**. The Peterson staining method was used to analyze the pollen viability[66]. For counting of pollen, three mature flower buds containing dehiscent anthers were collected and dipped in 13 μl Perterson staining solution (10% ethanol, 0.01% malachite green, 25% glycerol, 0.05% acid fuchsin, 0.005% orange G, 4% glacial acetic acid) for 10 s on a microscope slide, which was then covered by a cover-slip; for the staining of entire anthers, 8 non-dehiscent anthers from mature flower buds were dissected under a binocular, immersed in 30 μl Perterson staining solution on a microscope slide, and stained for overnight. Subsequently, slides were heated on a hotplate at 80 °C for 10 min (for pollen counting) or 60 min (for entire anther staining) to distinguish aborted and non-aborted pollen grains. Slides were analyzed and imaged using a light microscope.

**Reporting summary**. Further information on research design is available in the Nature Research Reporting Summary linked to this article.

## Data availability
The mass spectrometry proteomics data have been deposited to the ProteomeXchange Consortium via the PRIDE[67] partner repository with the dataset identifier PXD009959 (http://proteomecentral.proteomexchange.org/cgi/GetDataset?ID=PXD009959). The *Arabidopsis* mutants and transgenic lines, as well as plasmids generated in this study are freely available from the corresponding author upon request. The source data of the immunoblots of Figs. 2b, 4c, 9b and the data underlying Figs. 1c and 8b are provided in the Source Data file.

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

## Acknowledgements

The authors are grateful to Anne Harzen (Max Planck Institute for Plant Breeding Research) for technical assistance. The authors thank Maren Heese and Paul Larsen for critically reading the manuscript. We kindly acknowledge the Salk T-DNA collection, the GABI-Kat T-DNA collection, the *Arabidopsis* Biological Resource Center (ABRC) and the European *Arabidopsis* Stock Centre (NASC) for providing seeds of T-DNA lines used in this report. This work was supported by core funding of the University of Hamburg.

## Author contributions

C.Y. and A.S. conceived the experiments. C.Y., Y.H., K.S., and F.B. performed the experiments and statistical analyses; S.C.S. and H.N. performed the mass spectrometry experiment and data analysis. C.Y. and A.S. analyzed the data. C.Y. and A.S. wrote the manuscript.

## Additional information

**Competing interests:** The authors declare no competing interests.

