## [Peer Review File · Nature Communications]

Reviewers' comments:

Reviewer #1 (Remarks to the Author):

In the manuscript entitled "SWITCH 1/DYAD is a novel WINGS APART-LIKE antagonist that maintains sister chromatid cohesion in meiosis" Yang et al. report on the identification SWI1 as a new meiotic regulator of cohesin. They suggest that this protein antagonize the cohesin unloader WAPL and therefore, it is a functional homolog of Sororin in plants. SWI1 is localized to the chromatin in leptoten and disappear in zygoten. The authors show interaction between N-terminal SWI1 and N-terminal PDS5A by 2HYS and in vitro pull-down assay. They demonstrate that phosphorylation of SWI1 C-terminal by CDKA is essential for its activity in cells. While SWI1 has been known to be a meiotic gene involved in cohesion, its biological function was unknown. This study assigns a function to the gene and establishes that cohesion mechanism is an evolutionary conserved process. Most of the experiments support the conclusions. However, some concerns need to be addressed and some experiments need to be extended in order to provide a comprehensive characterization of the biological function and mechanism. Based on the scientific merit and novelty of the work, I suggest to consider accepting the manuscript after a major revision.

1. In general, the order of the figures does not follow the text. Reading is not flowing and requires to move back and forth between figures. For example, 1f is mentioned only after 3d. 3c appears in the text after 3d etc. Reorganization of the text is required.
2. Bioinformatics data should include sequence alignment of SWI1 from Arabidopsis, rice and maize. In addition, can the binding site be identified by sequence alignment or motif discovery (e.g. MEME- <http://meme-suite.org/tools/meme>) by using Sororin and SWI1 homologs? The similarities between Arabidopsis PDS5 paralogs should be discussed. Are they redundant?
3. The displacement experiment in figure 3d is not convincing. It seems that the amount of the displaced bound WAPL is not correlated with the concentration of SWI1, and separated WAPL1. In addition, the quality of the gel is poor. The K_d of SWI1-PDS5A and WAPL1-PDS5A be calculated by SPR or similar technique. Can the SWI1-PDS5 interaction be validated in cells? (co-IP, FRET). Does SWI1-PDS5 and WAPL1-PDS5 complexes co-exist?
4. SWI1 phosphorylation. The stability of the 9A and 13A protein and the effect of phosphomimic S/T->D mutation should be tested.

Reviewer #2 (Remarks to the Author):

Please find enclosed my critical evaluation of the manuscript "SWITCH 1/DYAD is a novel WINGS APART-LIKE antagonist that maintains sister chromatid cohesion in meiosis" submitted by Chao Yang et al. (Arp Schnittger laboratory).

The study ascribes a molecular function to the previously identified SWI1/DYAD factor in meiosis. This is remarkable for many reasons: First, since 2001 numerous studies described multiple features of corresponding mutant alleles but failed to place the protein in a known regulator network. Second, SWI1 turned out to be an antagonist of WAPL and therefore the study sheds light on cohesin regulation in higher eukaryotes. While WAPL antagonists have been known in vertebrates (and to a certain extent in fly), it was not clear if the conserved cohesin complex is regulated in a similar manner also in plants and further phyla/kingdoms. This study highlights that the general logic of the so-called "pro-phase pathway" is broadly conserved. Third, this study utilizes live cell imaging of meiosis which represents a major technical leap. Also, the amount of work, the extensive genetic epistasis analysis and the functional analysis of phosphorylation sites are impressive.

Nevertheless, some experiments need refinement or completion, some of the writing needs to be

extended and/or corrected (see detailed review below).

Most importantly, the study falls short in describing the key technology employed (live cell imaging of meiosis) and only gives a reference (Prusicki et al.,) of a submitted manuscript. Either the authors provide an in-depth description of this technique in this manuscript, or publication of this study has to be withheld until the other one is publicly available.

This also relates to certain key aspects of the analysis:

- How is meiotic progression staged, especially in mutant plants with different timing of hallmark features? This is most likely to be found in the unavailable "Prusicki et al." manuscript.
- The technical/functional description of the central molecular tool, a plant line expressing GFP-tagged REC8, cannot be found in literature or in this manuscript (...but most likely it is also in the "Prusicki et al." manuscript).
- The technical/functional description of a plant line expressing GFP-tagged ZYP1 is also not part of the given study (...but most likely also to be found in the "Prusicki et al." manuscript).

Please find more details below:

Abstract:

First sentence...pl rephrase: "cohesin" embraces, "cohesion" not.

General remark: the authors should take care, throughout the entire manuscript, when to use the term "cohesin" or the term "cohesion". Semantic precision is needed there, since monitoring e.g. REC8-GFP yields information about the presence/absence etc of the kleisin subunit of a cohesin complex, but does not necessarily yield information on actual sister chromatid cohesion.

Delete the term "class of" in the sentence that starts "Here we show that....". It is unclear if SWI1 truly defines a novel class of WAPL antagonists or is just another antagonist.....

Last sentence...pl rephrase: make two sentences. Speculating on convergent evolution seems reasonable but "linking mitosis and meiosis" appears unclear in this context. Either explain in more depth or skip.

Main text:

REC8-GFP reporter line not described. Pl provide information and data confirming its function (or, alternatively, a valid reference).

Figure 1a: why is there no data for wapl1 wapl2 double mutant for timepoints -24 to -8 hours? Please provide this data for comparison. How has the measurement and the quantification been performed? What has been measured, how has the data been compared and normalised? How have size/area differences been taken into account? Please explain.

General remark: please describe staging of meiotic events in mutant plant lines and provide evidence or a valid reference. Why is the ASY3-RFP only shown in some experiments? Has it been used for staging in all cases?

Suppl. Fig. 1: It is unclear if the WAPL protein is bound to chromatin or just residing in the nucleus. One possibility to differentiate would be to make chromatin spreads and probe for WAPL presence after washing away the nucleoplasm. Please perform this (or an alternative) experiment.

Please provide reference for "Peterson staining".

General remark: pl reorder results according to the flow of the story. E.g.: Figure 1 c, e and f are only mentioned in the text after introducing results from Figures 2 and 3.

Pl correct text: REC8-GFP does not allow monitoring cohesion defects....; SWI1 localisation may depend on cohesion, but to ask if it depends on cohesin makes more sense in the context.

The sentence "To address whether SWI1 localization also depends....etc" needs to be revisited: the logic is not correct.

Furthermore, the meiotic stages in swi1 mutant plants needs to be carefully assessed to support the author's statement. This needs a more thorough description of the employed live imaging and staging procedure (see above).

The authors claim that REC8-GFP is not properly localised to chromatin in swi1 mutants (Figures 1c and S6) yet the pictures (and videos) show that there is some texture of the staining. This, and also the minimal staining in metaphase I cells (Fig 1c) indicates that some REC8-GFP can bind to / is retained at chromatin. It would be interesting to see the quantification performed on chromatin spreads, not obscured by REC8-GFP in the nucleoplasm, to understand the quantitative effect of swi1 mutation(s) on REC8-GFP chromatin binding. The current set-up of experiments is not suited to yield this answer. Such an experiment would also allow to compare earlier results on swi1-2, claiming that REC8 was loaded, though with a different pattern (Mercier 2001/2003). It would also be interesting to compare different swi1 mutant alleles.

Furthermore, especially Fig 1c / Fig S6 / REC8-GFP in swi1 mutants appear to display different (earlier) meiotic stages and the authors need to explain extensively why they believe that they have captured the correct stages.

The same as above applies to the localisation of SWI1 (Fig 2c): how much is retained at the chromatin in the absence of REC8? The current set-up, with the nucleoplasm intact does not allow to address this question, yet a meiotic chromatin spread preparation would give a clear answer.

Please indicate the fertility (e.g. number of seeds) for the swi1 wapl1 wapl2 triple mutant in comparison to the swi1 single and wapl1 wapl2 double mutants and discuss the result. Fig. S2 does not provide pictures of siliques of the different swi1 alleles...pl complete.

Interaction of SWI1 with PDS5A:

Please provide original gels/blots of experiments (see also remarks relating to Figure S9) and an explanation of background bands.

Competition of SWI1 and WAPL for PDS5A binding:

Please include a control of SWI1 binding to beads alone;

Please provide original gels/blots of experiment (see also remarks relating to Figure S9) and explanation of background bands.

Please revise sentence "To this end, we loaded recombinant WAPL1-PDS5A heterodimers onto PDS5A bound beads...."....is this true? If yes, this needs explanation.

Furthermore, please explain experiment with more words: not only more WAPL1 is released with increasing SWI1 concentrations, also more SWI1 is bound to PDS5A....etc....

Please indicate for the SWI1 homologs OsAM1 and ZmAM1 that the interaction assays has been performed with "Arabidopsis" PDS5A.

SWI1 hypomorphic mutants

Please additionally perform immune-spreads to clarify if SWI1-GFP with mutated phospho-sites is retained at chromatin or stabilized in general.

As mentioned above, also the stages displayed in Figure 4c appear very early and not necessarily related to "zygotene" or "pachytene" stages. The authors may have good arguments but the need to share them.

Please provide information about plant fitness and fertility of plant lines carrying hypomorphic versions of SWI1.

Discussion:

Last sentence: please skip "...in both mitosis and meiosis", since the data in the manuscript exclusively relates to meiosis.

The reviewer misses an extensive discussion of the results with respect to the other kleisins present in plants. The meiotic spreads (not yet presented/performed) of SWI1 are of relevance here, since it is unclear if SWI1 is only associated to PDS5A / REC8. Fig. 2c is not conclusive and it might very well be that SWI1 localises to further PDS5/kleisin combinations.

Methods:

Plant lines: information on rec8 mutant line is missing...pl provide information

Protein expression and purification:

Fig. S9: a-c: please provide Western blots (entire gel) and MS data that verify the identity of all the protein bands on the CBB gels. Please provide explanation of the multiple bands or perform a more thorough protein purification. SDS without CDKA;1 is missing – please provide proof.

Fig. S9: d: this CBB gel is mysterious, since the band patterns, the loading scheme and the color-coded arrow heads do not fit to each other. E.g. lane 1: a CDKA;1 band is indicated with black arrowhead, but the loading scheme indicates only presence of SDS and SWI1 (1-300). Pl correct.

Chromosome spreading:

Pl provide all needed information for the "enzyme solution"....there are no concentrations indicated.

In vitro kinase assay:

Please provide details of the "slight" modifications.

Confocal microscopy:

Please provide valid reference (Prusicki et al., submitted) or detailed information (see above).

Some typos: coinciding; sporulation;

Use either "movie" or "video" when referring to the supplemental videos for consistency.

Detailed response to the reviewer comments on Yang et al.

Reviewer #1 (Remarks to the Author):

In the manuscript entitled “SWITCH 1/DYAD is a novel WINGS APART-LIKE antagonist that maintains sister chromatid cohesion in meiosis” Yang et al. report on the identification SWI1 as a new meiotic regulator of cohesin. They suggest that this protein antagonize the cohesin unloader WAPL and therefore, it is a functional homolog of Sororin in plants. SWI1 is localized to the chromatin in leptoten and disappear in zygoten. The authors show interaction between N-terminal SWI1 and N-terminal PDS5A by 2HYS and in vitro pull-down assay. They demonstrate that phosphorylation of SWI1 C-terminal by CDKA is essential for its activity in cells. While SWI1 has been known to be a meiotic gene involved in cohesion, its biological function was unknown. This study assigns a function to the gene and establish that cohesion mechanism is an evolutionary conserved process. Most of the experiments support the conclusions. However, some concerns need to be addressed and some experiments need to be extended in order to provide a comprehensive characterization of the biological function and mechanism. Based on the scientific merit and novelty of the work, I suggest to consider accepting the manuscript after a major revision.

We very much appreciate the positive feedback and like to thank this reviewer for his/her constructive comments.

1. In general, the order of the figures does not follow the text. Reading is not flowing and requires to move back and forth between figures. For example, 1f is mentioned only after 3d. 3c appears in the text after 3d etc. Reorganization of the text is required.

We have reorganized the text and the figures hoping that the flow of the arguments and data presented is better to follow now.

2. Bioinformatics data should include sequence alignment of SWI1 from arabidopsis, rice and maize. In addition, can the binding site be identified by sequence alignment or motif discovery (e.g. MEME- <http://meme-suite.org/tools/meme>) by using sororin and SWI1 homologs? The similarities between arabidopsis PDS5 paralogs should be discussed. Are they redundant?

When we compared SWI1 with Sororin proteins, we could not identify an obviously common domain. There are stretches of similar sequences. However, these are of low complexity and often contain gaps and hence are indistinguishable from random similarities found by any protein-protein comparison.

Likewise, we did an alignment of SWI1 homologs from different plant species by ClustalW2 including *Arabidopsis thaliana*, *Zea mays*, *Brassica rapa*, *Puccinellia tenuiflora*, *Cucumis sativus*, *Glycine max*, *Oryza sativa*, *Brachypodium*

distachyon, and *Sorghum bicolor*. We found three stretches of conserved amino acids in the N-terminal part of SWI1 homologs of which one or combination of them could be responsible for the binding with PDS5. We are happy to include this and/or a MEME figure of these domains in the manuscript. However, we have already 10 main and 16 supplementary figures and would like to keep this decision at the hands of this reviewer and the editor. To our understanding, such an alignment can be easily generated by the interested readers.

The sequence similarities of the Arabidopsis PDS5 protein family has been described by Pradillo and colleagues (2015) revealing that PDS5 paralogs are between 23 and 31% identical and between 20 and 39% similar to PDS5A. Moreover, there appears to be a high level of functional redundancy as also shown by Pradillo and colleagues.

Following the comment of this reviewer, we tested the interaction of SWI1 with the other PDS5 paralogs in Arabidopsis. We found that SWI1 weakly, yet reproducibly interacts with PDS5B and PDS5D, and strongly with PDS5C and PDS5E, suggesting that SWI1 is a common interaction partner of the PDS5 family. The data is presented in the revised Supplementary Fig. S9d.

3. The displacement experiment in figure 3d is not convincing. It seems that the amount of the displaced bound WAPL is not correlated with the concentration of SWI1, and separated WAPL1. In addition, the quality of the gel is poor.

We performed the western blot again and replaced the corresponding picture in the revised version hoping that the quality of this blot is convincing. The repetition produces the same result as before, i.e. the more SWI1 we add, the more WAPL is found in the supernatant (now Fig. 4c). Conversely, the more SWI1 we add, the more SWI is bound to PDS5.

To substantiate the competition of SWI1 with WAPL for binding to PDS5, we have added two more experimental sets: First, we performed a ratiometric BiFC assay that allows a semi-quantitative assessment of the binding affinities of PDS5A with SWI1 and WAPL1. This assay showed that SWI1 binds stronger to PDS5A than WAPL1 to PDS5A (Fig. 4a and b). Second, we performed an *in vivo* competitive binding experiment using tobacco cells. There, we added SWI1 to a BiFC assay between WAPL1 and PDS5A demonstrating that SWI1 outcompeted WAPL1. Conversely, the addition of WAPL1 could not block the interaction between SWI1 and PDS5A.

We think that the three assays together provide strong evidence that SWI1 antagonizes WAPL1.

The Kd of SWI1-PDS5A and WAPL1-PDS5A be calculated by SPR or similar technique.

We have tried to measure the Kd of SWI1-PDS5A and WAPL1-PDS5A using MicroScale Thermophoresis (MST) with a fluorescently labeled PDS5A (see image below). We got a Kd 2.1 and 7.1 μ M for SWI1-PDS5A and WAPL1-PDS5A, respectively, suggesting the interaction of SWI1-PDS5A is stronger than WAPL1-

PDS5A. However, we could not saturate the binding of PDS5A with both SWI1 and WAPL1 since both proteins start to aggregate at high concentrations in our buffer. This makes the K_d calculation less precise and therefore, we only attach the result here and did not add them in the manuscript.

Can the SWI1-PDS5 interaction be validated by in cells? (co-IP, FRET).

The fact that one flower bud contains only very few meiocytes (less than 1%) makes Co-IP experiments almost impossible. To provide further in vivo support for the SWI1-PDS5 interaction, we have used BIFC assays in tobacco leaf cells that confirmed the interaction of SWI1 with all PDS5 paralogs. The data is shown now in Fig. 3c, Supplementary Fig. S9d. With this, we provide now evidence for the interaction of SWI1 with PDS5 by three different assays, i.e. Y2H, GST-pull down and BIFC (Fig. 3).

Does SWI1-PDS5 and WAPL1-PDS5 complexes co-exist?

WAPL1 shows a very diffused localization pattern in early prophase and only starts to be enriched on chromatin at late prophase when SWI1 is released (see Fig. 1b). This is also confirmed by our immuno-localization results (Supplementary Fig. 1c). Therefore, while we cannot completely rule out that both complexes exist in parallel, it seems very likely that SWI1-PDS5 and WAPL1-PDS5 do not co-exist or at least that WAPL1-PDS5 complexes are not predominant when SWI1 is highly expressed in early prophase. We have included this hypothesis in the discussion part.

4. SWI1 phosphorylation. The stability of the 9A and 13A protein and the effect of phosphomimic S/T->D mutation should be tested.

We have generated transgenic plants expressing a phospho-mimetic SWI1 version in which we have replaced S or T in all 13 S/T-P sites with Aspartate. This phospho-mimicry version (SWI1-13D-GFP) showed a similar localization pattern as the wildtype consistent with the idea that phosphorylation is necessary for SWI1 removal. Unfortunately, we cannot fully rule out that the negative charges provided by our point mutations are not sufficient to trigger premature degradation (please note that phosphate groups carry more negative charges than Aspartate or Glutamate). None-the-less, these experiments gave rise to the hypothesis that phosphorylation alone is not sufficient for SWI1 turnover hinting at a higher order co-ordination of SWI1 removal. We have discussed this point in the revised text.

To elaborate further the control of SWI1 degradation, we performed cell free degradation assays. These assays revealed that a dephospho-mutant version of SWI1 is more stable than the wildtype SWI1 protein. Furthermore, we found evidence that SWI1 degradation is promoted by CDK phosphorylation since the addition of the Cdk inhibitor roscovitine enhanced the stability of SWI1 (See Fig. 9).

In parallel, we identified five putative destruction boxes in SWI1 and have generated plants in which we altered their consensus sites. First, we only deleted the two highly conserved sites. This led to no altered SWI1 localization pattern. However, when we inactivated all five putative APC/C recognition sites, we found that SWI1 was stabilized *in planta* (see Fig. 9).

Taken together, we think that we provided compelling evidence for the degradation of SWI1. This degradation appears to be dependent on CDK phosphorylation (we have shown in the version before that SWI1 is stabilized in hypomorphic *cdka;1* and *sds* mutants) and relies on the APC/C.

Reviewer #2 (Remarks to the Author):

Please find enclosed my critical evaluation of the manuscript “SWITCH 1/DYAD is a novel WINGS APART-LIKE antagonist that maintains sister chromatid cohesion in meiosis” submitted by Chao Yang et al. (Arp Schnittger laboratory).

The study ascribes a molecular function to the previously identified SWI1/DYAD factor in meiosis. This is remarkable for many reasons: First,

since 2001 numerous studies described multiple features of corresponding mutant alleles but failed to place the protein in a known regulator network. Second, SWI1 turned out to be an antagonist of WAPL and therefore the study sheds light on cohesin regulation in higher eukaryotes. While WAPL antagonists have been known in vertebrates (and to a certain extent in fly), it was not clear if the conserved cohesin complex is regulated in a similar manner also in plants and further phyla/kingdoms. This study highlights that the general logic of the so-called “pro-phase pathway” is broadly conserved. Third, this study utilizes live cell imaging of meiosis which represents a major technical leap. Also, the amount of work, the extensive genetic epistasis analysis and the functional analysis of phosphorylation sites are impressive.

We also like to thank this reviewer for his/her positive judgment of our work and the constructive comments to improve the manuscript.

Nevertheless, some experiments need refinement or completion, some of the writing needs to be extended and/or corrected (see detailed review below).

Most importantly, the study falls short in describing the key technology employed (live cell imaging of meiosis) and only gives a reference (Prusicki et al.,) of a submitted manuscript. Either the authors provide an in-depth description of this technique in this manuscript, or publication of this

study has to be withheld until the other one is publicly available. This also relates to certain key aspects of the analysis:

- **How is meiotic progression staged, especially in mutant plants with different timing of hallmark features? This is most likely to be found in the unavailable “Prusicki et al.” manuscript.**

The manuscript by Prusicki et al. that provides details on the live cell imaging has been accepted pending revision in eLife. We have uploaded the paper by Prusicki et al. on the Nature communication server and have also deposited a preprint version in bioRxiv. Thus, the paper is publically available and we hope that this information will provide sufficient information on how we have performed the live cell imaging and staged meiosis in this work.

- **The technical/functional description of the central molecular tool, a plant line expressing GFP-tagged REC8, cannot be found in literature or in this manuscript (...but most likely it is also in the “Prusicki et al.” manuscript).**

Yes, the functionality of REC8-GFP is documented in Prusicki et al.

- **The technical/functional description of a plant line expressing GFP-tagged ZYP1 is also not part of the given study (...but most likely also to be found in the “Prusicki et al.” manuscript).**

We have added the description of the ZYP1 marker in this manuscript. Please see Supplementary Fig.6 for details. However, the functional redundancy of the two *ZYP1* genes and the tight linkage of *ZYP1a* and *ZYP1b* (separated only by ~2 kb) make a functional evaluation of ZYP1b-GFP challenging. Please note that we therefore cannot make a statement about its functionality. However, the localization observed with this marker matches very well the previously published pattern revealed by immuno-localization. Moreover, the expression of ZYP1b-GFP has no dominant negative effect. Given that we only use this marker for a refined staging, we hope that the reviewer agrees that it is justified to use this reporter line. Please note that we can already draw the same principle conclusions by using the functional ASY3 reporter line that we present in this study as well.

Please find more details below:

Abstract:

First sentence...pl rephrase: "cohesin" embraces, "cohesion" not.

We thank the reviewer for this comment as it made us aware of a problem with the WORD auto-correction function, which wants to convert cohesin to cohesion. We have now carefully double-check the entire text and corrected this where necessary, including the sentence mentioned by the reviewer.

General remark: the authors should take care, throughout the entire manuscript, when to use the term “cohesin” or the term “cohesion”. Semantic precision is needed there, since monitoring e.g. REC8-GFP yields information about the presence/absence etc of the kleisin subunit of a cohesin complex, but does not necessarily yield information on actual sister chromatid cohesion.

We double-checked the entire text. Please see comment above.

Delete the term “class of” in the sentence that starts “Here we show that....”. It is unclear if SWI1 truly defines a novel class of WAPL antagonists or is just another antagonist.....

We agree and have deleted “class of”.

Last sentence...pl rephrase: make two sentences. Speculating on convergent evolution seems reasonable but “linking mitosis and meiosis” appears unclear in this context. Either explain in more depth or skip.

What we wanted to say was that CDK activity appears to be at the core of both meiosis and mitosis as in both cases, it is used to orchestrate progression through the respective cell division program. Sororin is marked by Cdk

phosphorylation for degradation in mitosis and SWI1 removal is controlled by Cdk action in meiosis. However, we are already at the limit of our abstract size and have deleted “linking mitosis and meiosis”.

Main text:

REC8-GFP reporter line not described. Pl provide information and data confirming its function (or, alternatively, a valid reference).

Detailed information can be found in Prusicki et al., please see comment above.

Figure 1a: why is there no data for *wapl1 wapl2* double mutant for timepoints -24 to -8 hours? Please provide this data for comparison. How has the measurement and the quantification been performed? What has been measured, how has the data been compared and normalised? How have size/area differences been taken into account? Please explain.

We have now added the data for -24 till -8h for *wapl1 wapl2* demonstrating that REC8 levels stay high throughout meiosis in *wapl* double mutants.

For quantification, we measured the REC8-GFP signal intensity in each meiocyte every hour using Fiji software. Please note that REC8 is exclusively present on chromosomes in both WT and *wapl1 wapl2* mutants. For possible size/area differences, we applied at least 7 zeta stacks spanning the volume of

the nucleus. The REC8-GFP signal was only captured from the section that had the largest nuclear diameter (7 to 9 μ m) and used for subsequent quantification. For this optical section, the background signal was obtained and subtracted at each time point. At least 20 meiocytes were measured for each time point. The relative intensities were then calculated with respect to the maximal REC8-GFP signal.

Regarding a possible normalization, we have tried to measure the fluorescence of a histone HTA1-RFP reporter that was simultaneously expressed with REC8-GFP. However, when looking only at the HTA1-RFP signal, it turned out that its intensity diminished over time (see graph below). This could be due to several reasons. First, the expression of HTA1 itself might not be constant during meiosis. Indeed, She et al. (2014) have shown that histone H1.1 is highly dynamic during plant reproduction and appears to be substantially remodeled in meiosis. Second, RFP is known to be less photo-stable than GFP possibly resulting in a fading signal through bleaching during our long observation times.

When we compare the results with and without normalization through HTA1-RFP, we obtain a very similar REC8 eviction patterns (see figure below). Given the doubts about the normalization through HTA1-RFP, we thought it is fair to leave this out rather than relying on a possibly not fully correct normalization procedure. We hope that the reviewer agrees that the presentation of the pure intrinsic accumulation patterns of REC8 is a sound procedure.

General remark: please describe staging of meiotic events in mutant plant lines and provide evidence or a valid reference. Why is the ASY3-RFP only shown in some experiments? Has it been used for staging in all cases?

For the staging of REC8-GFP in *wapl1 wapl2* mutants, we have used the cell shape of the meiocyte and nucleolus position. Both aspects of meiocytes have been used before for staging, please see Wang et al. (2004) and Stronghill et al. (2014). These criteria also match our recent landmark system based on live cell imaging (Prusicki et al. in revision). In addition, we have used ASY3, REC8 and ZYP1 for staging. These markers have not all together been applied to all experiments but only when needed to keep the workload reasonable.

Since chromosome axis formation is very compromised in *swi1* mutants, it is impossible to use ASY3-RFP for staging. It is also necessary to mention that unlike most other meiotic mutants, the nucleolus migration also has defects in *swi1* and thus, the staging of *swi1* mutants for live cell imaging was mainly judged by the cell shape of the meiocyte, the nuclei number of the tapetal cells, and the size of the flower bud.

Suppl. Fig. 1: It is unclear if the WAPL protein is bound to chromatin or just residing in the nucleus. One possibility to differentiate would be to make chromatin spreads and probe for WAPL presence after washing away the nucleoplasm. Please perform this (or an alternative) experiment.

To address this question, we first revisited the localization pattern of WAPL1-GFP by live cell imaging with shorter scanning intervals and higher laser power revealing a comprehensive localization study of WAPL1 throughout meiosis (presented now in Fig. 1b). We find that WAPL1 starts to accumulate on

chromatin (formation of foci/short stretches) at late leptotene/early zygotene, i.e., at a timepoint when *SWI1* expression declines (Fig. 1b Fig. 2).

In addition, we also performed immunolocalization experiments for WAPL1-GFP as requested by this reviewer (see Supplementary Fig. 1c). However, WAPL accumulation is very low in meiocytes, a finding that we became already aware of in our live cell imaging approach. In fact, WAPL is much lower expressed than SWI1-GFP and REC8-GFP. As the reviewer is likely aware of, this makes immunolocalization experiments very challenging. None-the-less, immunolocalization of WAPL confirmed the conclusions drawn by live cell imaging. A very faint and dotty localization of WAPL1-GFP was found in late G2 phase and chromatin association of WAPL1-GFP became more obvious at late leptotene/early zygotene, which persisted until metaphase I.

Please provide reference for “Peterson staining”.

We have added the reference.

General remark: pl reorder results according to the flow of the story. E.g.: Figure 1 c, e and f are only mentioned in the text after introducing results from Figures 2 and 3.

We have reordered the figures and hope that the readability is now enhanced.

Pl correct text: REC8-GFP does not allow monitoring cohesion defects....; SWI1 localisation may depend on cohesion, but to ask if it depends on cohesin makes more sense in the context.

We have corrected this. Please see our above answer on the use of cohesin versus cohesion.

The sentence “To address whether SWI1 localization also depends....etc” needs to be revisited: the logic is not correct.

We have corrected this and write now “To address whether SWI1 localization also depends on cohesin, we introgressed the *SWI1-GFP* reporter into *rec8* mutants.”

Furthermore, the meiotic stages in *swi1* mutant plants needs to be carefully assessed to support the author’s statement. This needs a more thorough description of the employed live imaging and staging procedure (see above).

We have done this. Please see our above comments on this point.

The authors claim that REC8-GFP is not properly localised to chromatin in *swi1* mutants (Figures 1c and S6) yet the pictures (and videos) show that there is some texture of the staining. This, and also the minimal staining in metaphase I cells (Fig 1c) indicates that some REC8-GFP can bind to / is retained at chromatin. It would be interesting to see the quantification performed on chromatin spreads, not obscured by REC8-GFP in the nucleoplasm, to understand the quantitative effect of *swi1* mutation(s) on REC8-GFP chromatin binding. The current set-up of experiments is not suited to yield this answer. Such an experiment would also allow to compare earlier results on *swi1-2*, claiming that REC8 was loaded, though with a different pattern (Mercier 2001/2003). It would also be interesting to compare different *swi1* mutant alleles.

As proposed by the reviewer, we have performed immunolocalization studies of REC8-GFP in wildtype versus *swi1-3* mutants (See Supplementary Fig. S7b). Consistent with our live cell imaging results, we found that REC8-GFP was still loaded onto and is associated with chromatin in *swi1-3* mutants. However, the REC8-GFP signal in *swi1-3* mutants is very fuzzy and failed to form into thread-like structures as seen in the wildtype. This observation is consistent with the previous conclusions made by Mercier and colleagues that REC8 is loaded onto the chromatin but failed to organize into normal axis in *swi1-2* mutants. Furthermore, our data showed that different *swi1* mutant alleles used in study had a similar fertility and cohesion defects (Supplementary Fig. S2, S7).

We feel that the difference with the wildtype are clearly visible and we hope that the reviewer agrees that a quantification of possible slight allele specific differences goes beyond the scope of this work.

Furthermore, especially Fig 1c / Fig S6 / REC8-GFP in *swi1* mutants appear to display different (earlier) meiotic stages and the authors need to explain extensively why they believe that they have captured the correct stages.

As mentioned above, the migration of nucleolus is affected in *swi1* mutants. Notably, the nucleolus can move a bit to one side of the nucleus, but often does not reach the corner (see figure 5b, supplementary figure 7a, and supplementary video 3). The reason for this is not clear. However, this behavior gives the impression that we have captured an earlier stage. Please note that the cell shape of the meiocyte and the nuclei number of the tapetel cells appear to be not affected in *swi1* mutants, which allowed us to roughly stage meiotic phases in the mutants.

The same as above applies to the localisation of SWI1 (Fig 2c): how much is retained at the chromatin in the absence of REC8? The current set-up, with the nucleoplasm intact does not allow to address this question, yet a meiotic chromatin spread preparation would give a clear answer.

We performed immunostaining and show the result in supplementary Fig. S8.

Consistent with our live cell imaging, we see that some SWI1 protein is still present on chromatin. As the reviewer is likely aware of, immuno detection is not quantitative and we also feel that an exact quantification is not necessary for our conclusions. Clearly, residual levels of SWI1 are present on chromatin in the absence of REC8 possibly hinting at a binding to other cohesin complexes not containing REC8 or a cohesin-independent chromatin association. We have discussed this point in the revised manuscript.

Please indicate the fertility (e.g. number of seeds) for the swi1 wapl1 wapl2 triple mutant in comparison to the swi1 single and wapl1 wapl2 double mutants and discuss the result.

We added the data as supplementary Fig. 14 and present them in the results section of the manuscript.

Fig. S2 does not provide pictures of siliques of the different swi1 alleles...pl complete.

We have added this information in panel b of Fig. S2.

Interaction of SWI1 with PDS5A:

Please provide original gels/blots of experiments (see also remarks relating to Figure S9) and an explanation of background bands.

Please see the following figures to assess the original blots. The “background bands” are usually due to protein fragmentation during the purification process. Based on our experience, it is very typical for proteins with large MW to obtain additional fragments besides the full-length protein. Nonetheless, the crude protein extracts after purification satisfied most of our experimental aims.

Competition of SWI1 and WAPL for PDS5A binding:

Please include a control of SWI1 binding to beads alone;

We have included this as panel 2 in Fig. S9.

Please provide original gels/blots of experiment (see also remarks relating to Figure S9) and explanation of background bands.

Please see the following figures depicting the original blots for explanation.

Please revise sentence "To this end, we loaded recombinant WAPL1-PDS5A heterodimers onto PDS5A bound beads...."is this true? If yes, this needs explanation.

This is correct. We co-expressed HisMBP-WAPL1 and HisGST-PDS5A in *E.coli* , and since WAPL1 interacts with PDS5A, we co-purified the WAPL1-PDS5A complexes using GST binding resin. This procedure is indicated in the methods section.

Furthermore, please explain experiment with more words: not only more WAPL1 is released with increasing SWI1 concentrations, also more SWI1 is bound to PDS5A....etc....

We have added this.

Please indicate for the SWI1 homologs OsAM1 and ZmAM1 that the interaction assays has been performed with “Arabidopsis” PDS5A.

We mention this in the results part of the manuscript.

SWI1 hypomorphic mutants. Please additionally perform immune-spreads to clarify if SWI1-GFP with mutated phospho-sites is retained at chromatin or stabilized in general.

In the previous version, we have shown the co-localization of SWI1^{13A}-GFP with REC8 at late prophase (Fig. 6c), suggesting the retention of SWI1^{13A}-GFP on

chromatin. To substantiate this based on the reviewer comment, we also performed immunostaining for SWI1^{13A}-GFP, and the results corroborated the preservation of SWI1^{13A}-GFP on chromatin (see Supplementary Fig. 8b).

As mentioned above, also the stages displayed in Figure 4c appear very early and not necessarily related to “zygotene” or “pachytene” stages. The authors may have good arguments but the need to share them.

As mentioned above, we always carefully evaluated the meiotic stage by combing different criteria. To provide further support for the correct staging, we now show the co-localization of SWI1 with REC8 at pachytene in the revised version.

Please provide information about plant fitness and fertility of plant lines carrying hypomorphic versions of SWI1.

We added the data in supplementary Fig. S10.

Discussion:

Last sentence: please skip “....in both mitosis and meiosis”, since the data in the manuscript exclusively relates to meiosis.

Since Sororin is known to be essential for the cohesion maintenance in mitosis in

vertebrates and *Drosophila*, and our study shows that SWI1 takes up a similar role in antagonizing WAPL in meiosis, we think that it is justified to say that the mechanism of prophase pathway regulation by WAPL inhibitors appears to be a general mechanism present in both mitosis and meiosis. However, since the discussion was rephrased to match the style of Nature communications (versus Nature cell biology), this sentence was also re-phrased.

The reviewer misses an extensive discussion of the results with respect to the other kleisins present in plants. The meiotic spreads (not yet presented/performed) of SWI1 are of relevance here, since it is unclear if SWI1 is only associated to PDS5A / REC8. Fig. 2c is not conclusive and it might very well be that SWI1 localises to further PDS5/kleisin combinations.

At this moment, we cannot finally conclude that SWI1 only binds to REC8 cohesin complexes. The retention of SWI1 in *rec8* mutants suggests that it might bind to other cohesin complexes. However, so far only SYN3/RAD21.2 has been found to act in meiosis and gametogenesis. Its role, however, remains obscure since it was found to localize to the nucleolus and not to chromatin. We have now included a discussion on the other kleisins present in plants, i.e., RAD21.1/SYN2, RAD21.2/SYN3 and RAD21.3/SYN4.

Regarding the question whether SWI1 is only associated with PDS5A/REC8, we provide now data showing that SWI1 interacts with all other PDS5 paralogs in *Arabidopsis*. While SWI1 weakly, yet reproducibly interacted

with PDS5B and PDS5D, it strongly bound to PDS5C and PDS5E (see Supplementary Fig. 9d).

Methods:

Plant lines: information on rec8 mutant line is missing...pl provide information

We have added this.

Protein expression and purification: Fig. S9: a-c: please provide Western blots (entire gel) and MS data that verify the identity of all the protein bands on the CBB gels. Please provide explanation of the multiple bands or perform a more thorough protein purification. SDS without CDKA;1 is missing – please provide proof.

We have included western blots for all the proteins used in this study. SDS without CDKA;1 was added as well.

The identity of HisMBP-SWI1¹⁻³⁰⁰ and HisMBP-SWI1³⁰¹⁻⁶³⁹ was confirmed both by western blot and by the MS analysis after the kinase assay. We attach here the MS confirmation report for the identity of HisGST-PDS5A¹⁻⁸⁰⁹ and HisMBP-WAPL1 with the coverage of 81.61% and 68.1%, respectively (see pictures below). CDKA;1 and SDS have been presented by Harashima and Schnittger (2012). We hope that the reviewer and editor agree that is not

feasible to cut out all bands on our protein gels and perform MS. We also think that this is not the current standard.

The major bands shown in the CBB gels are the relevant proteins. Please note that most affinity purification resins show some unspecific binding giving rise to some background bands. For instance, it is known that an unspecific protein with a mass slightly higher than 70kDa is usually co-purified when using Nickel-charged affinity resins. We have indicated this band in our images (supplementary Fig. S15). However, these background bands do not affect our experiments or alter any conclusion.

PDS5A

WAPL1

Fig. S9: d: this CBB gel is mysterious, since the band patterns, the loading scheme and the color-coded arrow heads do not fit to each other. E.g. lane 1: a CDKA;1 band is indicated with black arrowhead, but the loading scheme indicates only presence of SDS and SWI1 (1-300). PI correct.

We have corrected this.

Chromosome spreading: PI provide all needed information for the “enzyme solution”there are no concentrations indicated.

We added this information.

In vitro kinase assay: Please provide details of the “slight” modifications.

We have added this information and write now “...with slight modification using Strep-Tactin agarose (iba) instead of Ni-NTA agarose for the purification.”

Confocal microscopy: Please provide valid reference (Prusicki et al., submitted) or detailed information (see above).

We have provided this information (please see also above).

Some typos: coinciding; sporulation;

We have corrected the typos and double-checked the manuscript for proper orthography.

Use either “movie” or “video” when referring to the supplemental videos for consistency.

We have harmonized the nomenclature and only refer now to videos.

REVIEWERS' COMMENTS:

Reviewer #1 (Remarks to the Author):

I reviewed the revised version of the manuscript by Yang et al. The authors have adequately addressed my concerns and I support publishing the paper after a minor revision.

Minor points:

1. I accept the authors' response regarding the presentations of sequence alignment and motif discovery. However, I suggest that in the discussion the authors will briefly describe these results and address the structural basis of the interaction in the context of the known structural knowledge.
2. Figure 4b. Although the difference in affinities is unquestionable, the P value should be indicated in the legend.
3. Figure 9b. Indicate in the legend what in the control band shown in the CBB panel.
4. Figure 9c. If I understand correctly the experiment has been done twice. It is not acceptable to calculate SD based on 2 experiments. The 2 experiments should be shown side by side in a bar chart.
5. Figure 10. The cohesin model does not properly describe current knowledge of cohesin architecture. REC8 interaction with SMC3 is most likely similar to the corresponding interaction with RAD21 and involve the SMC3 coiled coil domain above the head.

Reviewer #2 (Remarks to the Author):

The authors addressed all concerns. Well done!

The study (and the revision) is very thorough, very extensive and very interesting!

REVIEWERS' COMMENTS:

Reviewer #1 (Remarks to the Author):

I reviewed the revised version of the manuscript by Yang et al. The authors have adequately addressed my concerns and I support publishing the paper after a minor revision.

We very much appreciate the support of this reviewer and would like to thank this reviewer once more for his/her constructive comments to improve our manuscript in the peer-review processes.

Minor points:

1. I accept the authors' response regarding the presentations of sequence alignment and motif discovery. However, I suggest that in the discussion the authors will briefly describe these results and address the structural basis of the interaction in the context of the known structural knowledge.

We have added a short paragraph on this point in the discussion and write now: "While a protein sequence-based alignment of the first 300 amino acids of the Arabidopsis SWI1 with its orthologs in other plant species including Brachypodium, bean, maize, sorghum, rapeseed, and rice revealed three conserved domains, no clear motif is emerging and further work will be required to address whether one of these domains or a specific combination mediates the interaction with PDS5. Including Sororin in this alignment did also not pinpoint to a PDS5 binding domain making it likely that the interaction between WAPL inhibitors and PDS5 is complex."

2. Figure 4b. Although the difference in affinities is unquestionable, the P value should be indicated in the legend.

We indicated the P value in the figure legend.

3. Figure 9b. Indicate in the legend what in the control band shown in the CBB panel.

We indicated the CBB panel in the figure legend.

4. Figure 9c. If I understand correctly the experiment has been done twice. It is not acceptable to calculate SD based on 2 experiments. The 2 experiments should be shown side by side in a bar chart.

We show now the two experiments in our figure.

5. Figure 10. The cohesin model does not properly describe current knowledge of cohesin architecture. REC8 interaction with SMC3 is most likely similar to the corresponding interaction with RAD21 and involve the SMC3 coiled coil domain above the head.

We corrected this.

Reviewer #2 (Remarks to the Author):

The authors addressed all concerns. Well done!

The study (and the revision) is very thorough, very extensive and very interesting!

We also like to thank this reviewer for the support and appreciate very much his/her constructive comments, which lead to a large improvement of the manuscript.